# Benchmarking Robustness of Text-Image Composed Retrieval

## Abstract

Text-image composed retrieval (TICR) aims to retrieve a target image through a query specified by an input image paired with text describing desired modifications to the input image. TICR has recently attracted attention for its ability to express precise adjustments to the input image–the query leveraging both information-rich images and concise natural language instructions. Despite their success, the robustness of popular TICR methodologies to either (1) real-world corruptions or (2) variations of the textual descriptions have never been evaluated systematically. In this paper, we perform the first robustness study of TICR, establishing three new diverse benchmarks for a systematic evaluation of robustness to corruption (in both the text and visual domains) in text-image composed retrieval, further probing textual understanding. For analysis of natural image corruption, we introduce two new large-scale benchmark datasets, CIRR-C and FashionIQ-C, for the open domains and fashion domains respectively–both of which feature 15 visual corruptions and 7 textual corruptions. Finally, to facilitate robust evaluation of textual understanding, we introduce a new diagnostic dataset CIRR-D by expanding the original raw CIRR dataset with synthetic data. CIRR-D's textual descriptions are carefully modified to better probe textual understanding across a range of factors: numerical variations, attribute variation, object removal, background variation, and fine-grained evaluation.

## 1 Introduction

Text-image composed retrieval (also known as composed image retrieval, or text-guided image retrieval) aims to retrieve an image of interest from a gallery of images through a composed query consisting of a reference image and its corresponding modified text. As a single word (e.g., 'dog') can correspond to thousands of images depicting dogs in various breeds, poses, and scenarios, the domain of natural language can be viewed as sparse and discrete, whilst the image domain is dense and continuous in contrast. Therefore, using *both* images and text together provides the advantage of expressing queries using the unique visual properties of each modality. Furthermore, the resulting retrieval requests are often much more semantically rich and nuanced than is possible by using images alone. This method holds potential in a variety of real-world applications, including fashion domain e-commerce Han et al. (2022b; 2023); Goenka et al. (2022); Han et al. (2022a); Chen et al. (2020) and open domain internet search Liu et al. (2021); Baldrati et al. (2022); Gu et al. (2023); Saito et al. (2023). However, existing text-image composed retrieval methods are primarily evaluated on clean data alone, while real world models often encounter distribution shifts Wang et al. (2021c), such as typos in the textual descriptions or/and image corruptions (e.g. weather changes). Furthermore, there is currently no analysis of whether such models utilize an understanding of the text as opposed to simply relying on correlations with the primary objects from the image query counterpart as a shortcut. For example, with a source image of a dog and its modified text 'change to two dogs on the table', the model might retrieve the target image by merely recognizing the words 'dog' and 'table' without actually performing any numerical counting. Whether text-image composed retrieval models are robust in real-world applications, where natural corruption exists in both images and text, remains unexplored. Additionally, the question of whether these models are robust across diverse textual understanding requirements is often not considered.

In this work, we take the first step towards providing a thorough evaluation of the robustness of text-image composed retrieval by building three new large-scale robustness benchmarks on both fashion and open

domains. We study the following two questions: **Q1:** *How robust are text-image composed retrieval models to natural corruption in both the visual and textual domains?* and **Q2:** *How robust are text-image composed retrieval models at textual understanding?*

To address the first question, we introduce two benchmark datasets on the text-image composed retrieval task. We propose our two benchmark datasets: FashionIQ-C and CIRR-C. These are based on the FashionIQ Wu et al. (2021) and CIRR Liu et al. (2021) datasets in the fashion and open domains respectively. Both datasets incorporate 15 visual corruptions and 7 textual corruptions, providing a comprehensive evaluation of model robustness against natural corruptions in both images and text. To answer the second question, we introduce a new diagnostic dataset CIRR-D to probe text understanding abilities across five fundamental scenarios: numerical variation, attribute manipulation, object removal, background variation, and fine-grained variation. In detail, the diagnostic dataset is constructed through the generation of synthetic triplets via image editing using a reference image and various text captions. This includes variations of both main captions and extended captions of the existing CIRR validation set. Our experiments show that the new benchmarks we introduce are suitable for robustness analysis against natural corruption on both image and text, as well as for probing text understanding abilities.

Our contributions are as follows: (1) We pioneer the analysis of the robustness of text-image composed retrieval methods against natural corruption (in both the visual and textual domains) and understanding under five categories of textual variation. (2) We introduce three new large-scale benchmarks including two benchmark datasets (FashionIQ-C and CIRR-C) to evaluate robustness against natural corruption in both image and text, and one diagnostic benchmark CIRR-D to probe text understanding robustness. (3) We provide an empirical analysis and conduct extended experiments to validate our findings: 1) model pretrained on large datasets will lead to better robustness, 2) text features from aligned space can help boost the robustness, while text features from independent space will reduce robustness 3) a modified text is more likely to enhance the model's discriminative ability when it minimizes the number of feasible targets and will harm performance when producing more candidate responses.

## 2 Related Works

**Robustness analysis.** The quantification of robustness aims to evaluate models' ability to defend against natural corruption Hendrycks & Dietterich (2019); Chantry et al. (2022); Wang et al. (2021b), or adversarial attacks Croce et al. (2020); Wang et al. (2021a;b), or to probe certain model ability such as logical reasoning Sanyal et al. (2022) or visual content manipulation Li et al. (2020a). Traditional works analyzing robustness mainly focus on a single modality. Exisiting work has often addressed either visual modality-based tasks (like image classification Hendrycks & Dietterich (2019) or face detection Dooley et al. (2022)), textual modality based tasks like text classification Zeng et al. (2021), or audio modality based task such as speech recognition Mitra et al. (2017). Recently, robustness analysis for multimodal tasks (which is closer to real life and attempts to take a step towards a reliable system) has appeared, but is still in its infancy. For example, Li et al. Li et al. (2020a) take the first step to systematically analyze the robustness of the multimodal task of Visual Question Answering (VQA) against 4 generic robustness measures including linguistic variation and visual content manipulation. However, it is limited to VQA tasks and doesn't introduce benchmarks to pinpoint sophisticated reasoning abilities. Schiappa et al. Chantry et al. (2022) introduce natural corrupted visual and textual benchmarks on text-to-video retrieval. However, the robustness analysis of the multimodal underlying hypothesis, which aims to generalize textual semantic and reasoning ability to visual space, is not discussed. In contrast, we consider the analysis of natural corruption to both image and text–further studying the underlying model textual understanding–and take the first step to conduct an extensive analysis of robustness of deep neural networks in text-image composed retrieval.

**Diagnostic analysis.** Recently, a range of benchmarks for visual understanding have been proposed, including datasets for image captioning Shekhar et al. (2017), visual question answering Johnson et al. (2017), visual reasoning Zerroug et al. (2022) and visio-linguistic compositional reasoning Thrush et al. (2022); Yuksekgonul et al. (2022); Ma et al. (2023). For text-image composed retrieval, the benchmarks can be categorized into sythetic-based datasets by cubes Vo et al. (2019) or natural scenes Gu et al. (2023), fashion-based datasets Han et al. (2017); Berg et al. (2010); Wu et al. (2021), object-state datasets Isola et al.

(2015) and open domain datasets Liu et al. (2021). Among them, the majority of the textual descriptions are limited to predefined attributes Han et al. (2017); Vo et al. (2019); Isola et al. (2015). To overcome this limitation, FashionIQ Wu et al. (2021) and CIRR Liu et al. (2021) leverage the flexibility of natural language, producing the most widely used benchmarks in the fashion domain and open domain respectively. We expand and categorize the validation set of the CIRR benchmark using both its main annotation and previously unused extended annotations. This expansion allows us to probe specific text understanding in five fundamental scenarios: numerical variation, attribute variation, object removal, background variation, and fine-grained variation. Similar to ours, many diagnostic datasets are also synthetic. CLVER Johnson et al. (2017) is a synthetic dataset to probe elementary vision reasoning including color, shape, and spatial relationships. CVR Zerroug et al. (2022) generates irregular shape, location, color, etc, and is designed for detecting the outliers from a small set of generated images. However, these are all simulated images and are not generated by imitating natural scenes. CasualVQA Agarwal et al. (2020) and CompoDiff Gu et al. (2023) generate images imitating natural scenes. However, CasualVQA is designed for visual question-answering tasks and the generated images include noticeable artifacts. While ComoDiff is designed for text-image composed retrieval, it generates images by replacing only the objects (noun) rather than attributes (numerical, adjectival, manipulation instructions) like ours. As a result, it cannot precisely pinpoint the target reasoning abilities. To assess the compositional abilities, visio-linguistic compositional reasoning diagnostic benchmarks introduce variations in objects Ma et al. (2023); Yuksekgonul et al. (2022), attributes Ma et al. (2023); Yuksekgonul et al. (2022), or text order Yuksekgonul et al. (2022); Thrush et al. (2022). However, these benchmarks only provide image-text pairs for single-modality queries rather than image-text-image triplets for multi-modality queries. Additionally, their text composition Ma et al. (2023); Yuksekgonul et al. (2022) may lack corresponding images, such as in the case of 'grass eat horse'. In comparison, our CIRR-D introduces the first visio-linguistic compositional reasoning benchmark tailored for text-image composed retrieval tasks in natural scenes.

**Text-image composed retrieval.** Composed image retrieval aims to retrieve the target image, where the input query is specified in the form of an image plus other interactions, such as relative attributes Parikh & Grauman (2011), natural language Chen et al. (2020); Vo et al. (2019), and spatial layout Mai et al. (2017), to describe the desired modifications. Among these, natural language has emerged as the most pervasive way for humans to convey intricate specifications to computers, resulting in increased attention, which has often led 'composed image retrieval' to become interchangeable with 'text-guided image retrieval' in the literature. To provide clarity on the composition of the query, we refer to this task as text-image composed retrieval. Traditional text-image composed retrieval models implement separate independent image and text encoders, whose features are combined with late fusion. For example, TIRG Vo et al. (2019) and Artemis Delmas et al. (2022) implement separate pre-trained ResNet as the image encoder and LSTMs as the text encoder. Through the power of CLIP's unified multimodal space Radford et al. (2021), recent text-image composed retrieval models achieved a noticeable improvement. For example, CLIP4CIR Baldrati et al. (2022) implements a light adapter as image-text late fusion and further tune it in target domains. Further based on CLIP, FAME Han et al. (2023) and CASE Levy et al. (2023) separately implement early cross attention between text and image, which shows obvious improvement.

## 3 Robustness Criteria for Text-Image Composed Retrieval

**Foundation of text-image composed retrieval.** Given a reference image $I_r$ and modified text $T_m$ as input query, the aim of text-image composed retrieval is to retrieve the target image $I_t$ from the gallery set $\{I_t^n\}_{n=1}^N$, where $N$ is the number of images in the gallery set. In other words, text-image composed retrieval aims to retrieve the target visual content through dense continuous images guided by sparse discrete text. We discuss the concept of 'dense' and 'sparse' in the semantic space, where a single semantic word can correspond to thousands of images. Therefore, text-image composed query can overcome the limitations of singular modality image retrieval, where text-image retrieval suffers from the imprecise descriptions and unlimited correct targets, and image-image retrieval suffers from expression limitation without the ability to generalize to different visual content. In light of this, the foundational abilities of text-image composed retrieval are threefold: (1) Image representation to provide a precise anchor in the dense continuous visual space; (2) Text representation to provide subtle or significant differences between various visual contents, providing

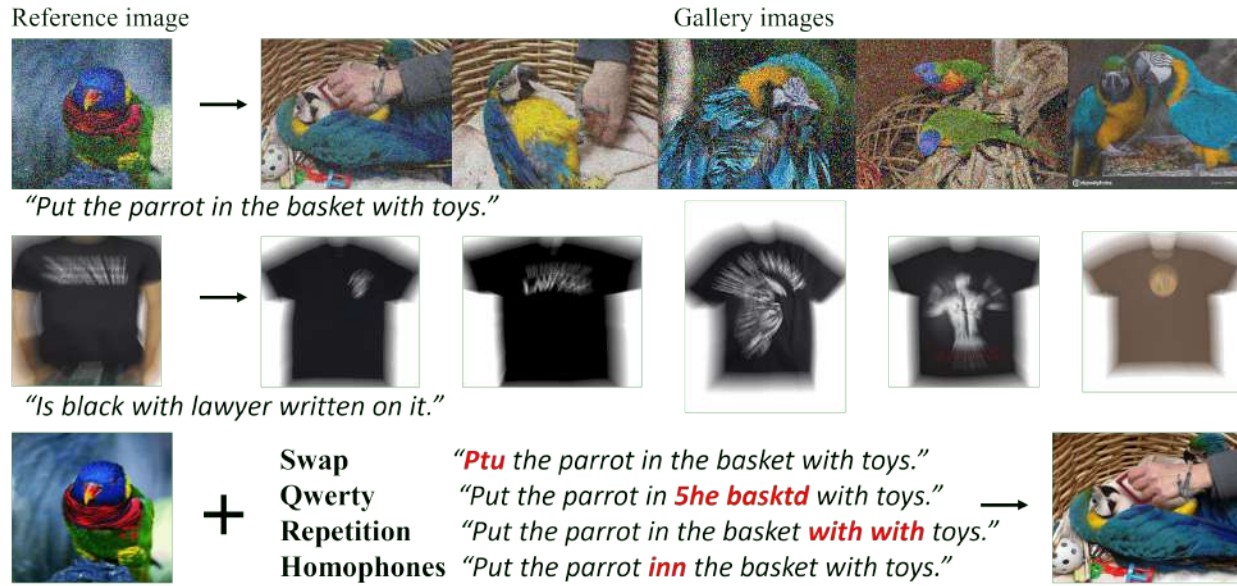

Figure 1: Sample visualization of proposed benchmarks under natural corruption with both visual and textual. **Top:** CIRR-C with impluse noise image corruption **Middle:** FashionIQ-C with zoom blur image corruption; **Bottom:** CIRR-C under character-level (Swap and Qwerty) and word-level (Repetition and Homophones) textual corruptions. Gallery images are shown without particular order.

an unprecise target direction the model can generalize to; (3) Generalize sparse modified text attributes to dense reference images to precisely predict the target visual content through the fusion of the vision and text modalities.

**Definition of robustness in text-image composed retrieval.** According to the foundation of text-image composed retrieval above, a robust model should demonstrate stable image feature extraction, text feature extraction, and modality fusion. In light of this, the robustness of text-image composed retrieval can be defined in twofold: *robustness against natural corruption* for both text and image and *robustness against textual understanding* for consistent reasoning between textual and visual modalities. Specifically, for robustness against natural corruption, we evaluate text-image composed retrieval models under ubiquitous corruptions frequently encountered in real-life scenarios in both visual and textual domains. The evaluation involves 15 standard image corruptions categorized into noise, blur, weather, and digital following Hendrycks & Dietterich (2019). Additionally, we evaluate 7 text corruptions categorized into character-level and word-level variations. Furthermore, for robustness against textual understanding, we evaluate common linguistic reasoning by selecting modified text with specific keywords or gallery set, categorized into numerical variation, attribute variation, object removal, background variation, and fine-grained variation, respectively.

**Evaluation metrics.** To evaluate the performance of models in text-image composed retrieval, we adopt the standard evaluation metric in retrieval, namely Recall@K denoted by R@K for short. Further, to measure robustness, we adopt relative robustness metrics $\gamma = 1 - (R_c - R_p)/R_c$ following the previous works Chantry et al. (2022); Hendrycks & Dietterich (2019), where $R_c$ and $R_p$ are the R@K under clean data and corrupted data, respectively. Additionally, in order to facilitate fair comparison among different models, we expand the work of Delmas et al. (2022) and established a unified testing platform for the convenient integration of various models. In detail, we set the gallery as the whole validation set as in Delmas et al. (2022); Baldrati et al. (2022), which includes more distractors and results in a larger need for discrimination, instead of setting the gallery the same as the query set as in Chen et al. (2020); Lee et al. (2021). Specifically for evaluating the fashionIQ dataset, we combine the two captions in a single query as Baldrati et al. (2022); Dodds et al. (2020) instead of combining the two modified captions in forward and reverse direction as Lee et al. (2021).

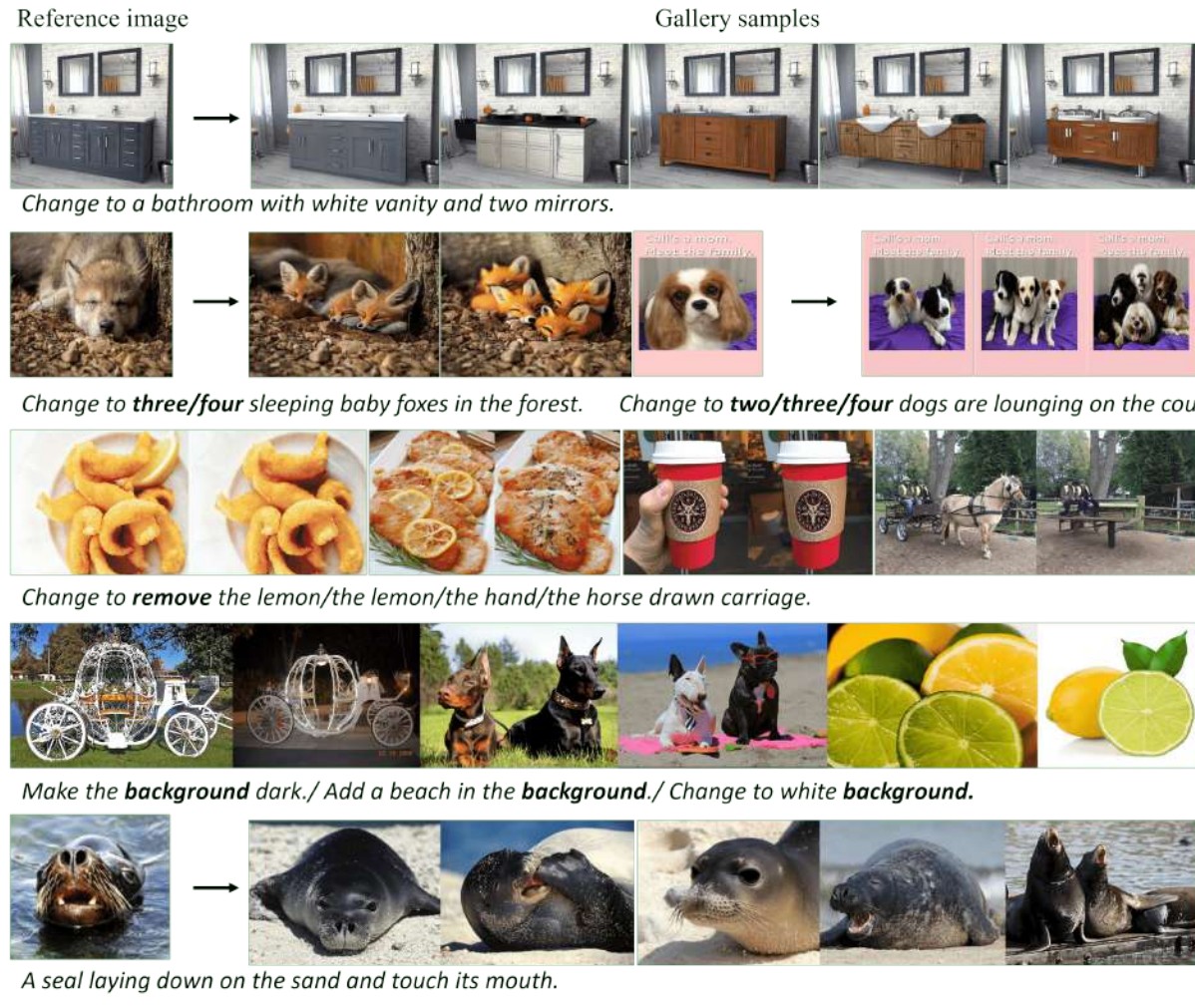

Figure 2: Visualization of samples from the proposed benchmarks probing textual understanding. Images from: **Row 1:** CIRR-D with attribute variations; **Row 2:** CIRR-D with number variations; **Row 3:** CIRR-D with object removal. **Row 4:** CIRR-D with background variations; **Row 5:** CIRR-D with fine-grained variations.

All the evaluated models are trained in three categories jointly and tested individually for dress, shirt, and toptee categories. The reported results for fashionIQ are the average of the three categories.

**Evaluation datasets.** We utilize three new benchmarks for our text-image composed retrieval experiments, which are generated based on two existing datasets: FashionIQ Wu et al. (2021) in the fashion domain and CIRR Liu et al. (2021) in the open domain. Both datasets include human-generated captions that distinguish image pairs. FashionIQ is based on the fashion domain containing 77,684 garment images, which can be divided into three categories: dress, shirt, and toptee. Each image in FashionIQ contains a single subject positioned centrally with a clean background. CIRR is composed of 21,552 real-life images extracted from NLVR2 Suhr et al. (2018), which contains rich visual content in diverse backgrounds. As shown in Figure. 1, we build our benchmark and evaluate text-image composed retrieval models on text and image natural corruption robustness. Further, as shown in Figure. 2, we expand the CIRR dataset and evaluate text understanding robustness. The creation details of our dataset can be found in supplementary A and more visualizations of these three benchmarks can be found in the supplementary B.

Table 1: Details of modality fusion for the evaluated models in this study. $R_i, M_t, T_i$ represent reference image, modified text and target image feature respectively. $C$ donates the composed of the two features. $<,>$ represents cosine similarity. The compared methods include: (1) without triplets supervision (above dash line): Image-only, Text-only, Pic2word Saito et al. (2023) and SEARLE Baldrati et al. (2023); (2) with triplets supervision (below dash line): TIRG Vo et al. (2019), MAAF Dodds et al. (2020), ARTEMIS Delmas et al. (2022), CIRPLANT Liu et al. (2021), CLIP4CIR Baldrati et al. (2022), FashionViL Han et al. (2022b).

| Model | Image encoder | Text encoder | $C_{RiMt}$ | $C_{RiTi}$ | $C_{MtTi}$ | Distance |
|---|---|---|---|---|---|---|
| Image-only (RN50) | ResNet50 | - | - | - | - | $<Ri, T_i>$ |
| Image-only (CLIP) | ResNet50x4 | - | - | - | - | $<Ri, T_i>$ |
| Text-only (CLIP) | - | GPT-2 Radford et al. (2019) | - | - | - | $<Mt, T_i>$ |
| Pic2word | ViT-L/14 | GPT-2 Radford et al. (2019) | textual inversion | - | - | $<C_{RiMt}, T_i>$ |
| SEARLE | ViT-L/14 | GPT-2 Radford et al. (2019) | textual inversion | - | - | $<C_{RiMt}, T_i>$ |
| TIRG | ResNet50 | LSTM | cat+residual | - | - | $<C_{RiMt}, T_i>$ |
| MAAF | ResNet50 | LSTM | self attn+ cross attn | - | - | $<C_{RiMt}, T_i>$ |
| Artemis | ResNet50 | LSTM | dot product | dot product | dot product | $<C_{RiMt},C_{TiMt}> + <C_{TiMt},M_t>$ |
| CIRPLANT | ResNet152 | BERT Devlin et al. (2018) | OSCAR Li et al. (2020b) | - | - | $<C_{RiMt}, T_i>$ |
| CLIP4CIR | ResNet50x4 | GPT-2 Radford et al. (2019) | cat + residual | - | - | $<C_{RiMt}, T_i>$ |
| FashionViL | ConvNet Huang et al. (2020) | BERT Devlin et al. (2018) | transformer | - | - | $<C_{RiMt}, T_i>$ |

*Natural vision and text corruption.* To evaluate the robustness of the text-image composed retrieval model against natural corruption in both image and text, we create our robustness benchmark CIRR-C and FashionIQ-C with 15 visual corruptions and 7 textual corruptions. For vision corruption, we follow Hendrycks & Dietterich (2019) to implement 15 standard natural corruptions which fall into four categories: noise, blur, weather, and digital, each having a severity from 1 to 5. For text corruption, we follow Rychalska et al. (2019) and implement the most related seven corruptions including four character-level corruptions and three word-level corruptions respectively.

*Diagnostic dataset.* Following the current methods Liu et al. (2021); Baldrati et al. (2022) reporting the results on the validation set, we expand and build our probing datasets CIRR-D based on the validation set of CIRR to pinpoint text understanding ability. We hypothesize that the model's corresponding reasoning capabilities can be evaluated when the modified text involves descriptions such as numbers, attributes, objects removal, or changing the background; and the ability to deal with fine-grained variations can be evaluated when the gallery images are highly similar following Liu et al. (2021). In light of this, we build the triplets (reference image, modified text, and target image) for our diagnostic dataset according to the appearances of specific keywords in the modified text: "zero" to "ten", "number" for numerical query; color, shape and size for attribute query, "remove" for object removal query; "background" for background variation query. The detailed statistics of the CIRR-D can be found in Table 5. The construction of CIRR-D dataset involves five probing categories from three sources as follows: (1) Existing Validation Set of CIRR: Comprising 2297 images and 4181 triplets, this set is widely utilized. Each image includes a subset of 6 highly similar images as a gallery to enhance the detection of fine-grained discriminative ability. (2) Auxiliary captions of the CIRR validation set: Although supplied, these captions have not been used in the conventional evaluations. These captions highlight differences in removed content or background changes between image pairs, but they may not provide sufficient information to precisely locate the target image. Consequently, we manually eliminated triplets that resulted in an excessive number of target images. (3) Synthetic image generation through Visual ChatGPT Wu et al. (2023): To integrate language reasoning with visual recognition, we augment the validation set by generating images through the process of image editing. The augmentation of the current distribution incorporates diverse variations in object quantity, color, shape, size, and existence. This can be regarded as a natural distribution shift occurring in real-world scenarios. To initiate this process, image captions for the CIRR validation set are generated by Visual ChatGPT. Subsequently, we create ten variants of the captions using ChatGPT, including four for numerical variants, three for color variants, two for size variants, and one for object removal. Afterward, leveraging the reference image and caption variants, Visual ChatGPT utilizes groundingDINO Liu et al. (2023) for object detection, segment anything Kirillov et al. (2023) for mask generation and stable diffusion Rombach et al. (2021) for target image generation. We manually remove implausible generated images from the synthetic dataset, ensuring that our synthetic image pairs maintain the original background and only modifies the specific areas mentioned in the text.

**Evaluated models.** We conduct experiments on six text-image composed retrieval models. The modality fusion of these approaches is summarized in supplementary Table 1 including Vo et al. (2019); Liu et al.

(2021); Baldrati et al. (2022); Delmas et al. (2022); Han et al. (2022b); Dodds et al. (2020), which can be categorized into the following overlapping categories: (1) Large pretrained models: FashionViL, CIRPLANT, CLIP4CIR, whose pretrained dataset size are 1.35 million, 6.5 million, and 400 million image-text pairs respectively; (2) Multi-task model: FashionViL, which is pretrained with four tasks simultaneously; (3) Light attention-based methods: ARTEMIS; (4) Transformer-based models: MAAF, CIRPLANT, CIRPLANT and FashionViL; (5) Lightweight models: MAAF, TIRG, and ARTEMIS (all with ResNet50 image encoder and LSTM text encoder for fair comparison). (6) Zero-shot models: Pic2word and SEARLE, which are pretrained on large-scale datasets and fine-tuned on the image-text pairs. We additionally design some methods with single-modality queries to gain better insights: (7) Single-modality models: (i) Image-only (RN50): Queried with images embedded by ResNet50, the same as evaluated TIRG, MAAF, ARTEMIS. (ii) Image-only (CLIP): Queried with images embedded by CLIP image encoder RN50x4, the same as evaluated CLIP4CIR. (iii) Text-only: Queried with text embedded using the CLIP text encoder. These methods were selected because the reproduced results match originally reported results. We test FashionViL Han et al. (2022b) in the fashion domain, CIRPLANT in the open domain, and all the remaining published models in both the fashion domain and open domains.

**Evaluation settings.** To ensure the fairness of the evaluation, we establish a standardized testbed for various models, excluding FashionViL, to unify the evaluation process. Additionally, to reproduce the original performance, we implement the official pre-trained weights for FashionViL and CLIP4CIR. We retrain the models and achieve similar results as reported for MAAF, TIRG, ARTEMIS, and CIRPLANT. In detail, we extend the existing ARTEMIS code framework to provide a convenient interface of different trained models, where TIRG and ARTEMIS are already implemented. For the input images for these models, CIRPLANT is based on frozen ResNet152 pre-trained features while other models take raw images as input. We implement a frozen ResNet152 image encoder so that we can introduce corruption to the raw image directly. For all the 15 image corruptions, we apply the highest severity of corruption to observe obvious performance differences. Regarding text input among these models, TIRG, MAAF, and ARTEMIS build the vocabulary based on appearance words in the target evaluation caption. In contrast, FashionViL, CIRPLANT, and CLIP4CIR implement their vocabulary from a large pretraining dataset. We implement textual corruption by directly modifying raw text. For a concise presentation, we report the results of FashionIQ by showing the average of the three categories: dress, shirt, and toptee.

**Implementation details** We supply 7 models (Pic2word, SEARLE, TIRG, MAAF, ARTEMIS, CIRPLANT and CLIP4CIR) in the same testbed to have a fair comparison with different benchmark datasets. The selection of models or datasets can be easily accomplished through input parameters. Further models can be implemented in our testbed by simply providing model structure files with the necessary interface. In detail, the necessary interfaces include image feature extraction, text feature extraction, feature composing process and distance comparison. Our testbed is currently compatible with two environments and seven models. CIRPLANT is implemented with Python(3.1) and Pytorch(1.8.1). Pic2word, SEARLE, TIRG, MAAF, CLIP4CIR and ARTEMIS were implemented with Python(3.8) and Pytorch(2.0). All the experiments are conducted and tested on NVIDIA A100 GPUs. We will maintain our code for benchmarking and testbed open source.

## 4 Results and Analysis

### 4.1 Natural corruption analysis

To evaluate whether the text-image composed retrieval models are robust under natural corruptions, we conduct experiments involving 15 visual corruptions, further categorized into **noise**, **blur**, **weather**, and **digital** corruptions on both the fashion domain and open domain. Table 2 presents the relative robustness $\gamma$ under the highest severity of each natural visual corruption, which shares the same trend across other corruption severities. To evaluate the robustness against textual corruption, experiments are conducted under 7 textual corruptions categorized as **character-level** and **word-level**. More experimental results can be found in the supplementary C.

Table 2: Relative robustness score for text-image composed retrieval under 15 natural image corruptions in CIRR-C Recall@10, FashionIQ-C Recall@10 and CIRCO-C mAP@10. Recall@10 and mAP@10 performance under clean conditions on the left. **Bold** is the highest relative robustness for the compared composed retrieval methods: Pic2word Saito et al. (2023), SEARLE Baldrati et al. (2023), TIRG Vo et al. (2019), MAAF Dodds et al. (2020), ARTEMIS Delmas et al. (2022), FashionViL Han et al. (2022b), CIRPLANT Liu et al. (2021), CLIP4CIR Baldrati et al. (2022) and BLIP2-CIR Li et al. (2023) following CLIP4CIR training process.

| | | Noise | | | Blur | | | | Weather | | | | Digital | | | |
|---|---|---|---|---|---|---|---|---|---|---|---|---|---|---|---|---|
| **CIRR-C** | Clean | Gauss. | Shot | Implu. | Defoc. | Glass | Motion | Zoom | Snow | Frost | Fog | Bright | Contr. | Elast. | Pixel | JPEG |
| Image-only(RN50) | 50.4 | 0.57 | 0.55 | 0.58 | 0.68 | 0.28 | 0.82 | 0.45 | 0.38 | 0.34 | 0.64 | 0.86 | 0.20 | 0.48 | 0.76 | 0.88 |
| Image-only(CLIP) | 36.2 | 0.56 | 0.55 | 0.58 | 0.66 | 0.32 | 0.83 | 0.49 | 0.52 | 0.45 | 0.77 | 0.91 | 0.24 | 0.41 | 0.78 | 0.91 |
| Text-only(CLIP) | 51.2 | 0.79 | 0.76 | 0.81 | 0.85 | 0.29 | 1.0 | 0.55 | 0.65 | 0.70 | 0.89 | 1.0 | 0.19 | 0.40 | 0.96 | 1.0 |
| Pic2word | 64.8 | 0.89 | 0.89 | 0.89 | 0.88 | 0.64 | 0.96 | 0.71 | 0.85 | 0.77 | 0.92 | 0.93 | 0.48 | 0.76 | 0.95 | 0.93 |
| SEARLE | 67.2 | 0.89 | 0.89 | 0.88 | 0.91 | 0.66 | 0.98 | 0.71 | 0.85 | 0.79 | 0.93 | 0.97 | 0.43 | 0.77 | 0.98 | 0.97 |
| TIRG | 55.1 | 0.34 | 0.36 | 0.34 | 0.48 | 0.21 | 0.70 | 0.43 | 0.31 | 0.22 | 0.40 | 0.70 | 0.12 | 0.47 | 0.74 | 0.84 |
| MAAF | 49.9 | 0.50 | 0.49 | 0.50 | 0.62 | 0.26 | 0.80 | 0.41 | 0.36 | 0.31 | 0.50 | 0.74 | 0.11 | 0.48 | 0.83 | 0.87 |
| ARTEMIS | 59.0 | 0.39 | 0.42 | 0.38 | 0.51 | 0.25 | 0.70 | 0.44 | 0.31 | 0.26 | 0.45 | 0.71 | 0.10 | 0.47 | 0.75 | 0.86 |
| CIRPLANT | 68.8 | 0.70 | 0.69 | 0.71 | 0.77 | 0.28 | 0.89 | 0.51 | 0.44 | 0.43 | 0.66 | 0.88 | 0.17 | 0.56 | 0.85 | 0.92 |
| CLIP4CIR | 80.3 | 0.68 | 0.68 | 0.69 | 0.77 | 0.28 | 0.90 | 0.52 | 0.55 | 0.60 | 0.80 | 0.91 | 0.16 | 0.39 | 0.91 | 0.92 |
| CLIP4CIR-L14-LAION400M | 66.6 | 0.80 | 0.81 | 0.82 | 0.92 | 0.59 | 0.96 | 0.61 | 0.74 | 0.73 | 0.94 | 0.97 | 0.34 | 0.72 | 0.95 | 0.93 |
| CLIP4CIR-L14-LAION2B | 81.7 | 0.79 | 0.80 | 0.81 | 0.92 | 0.57 | 0.96 | 0.64 | 0.84 | 0.77 | 0.94 | 0.97 | 0.33 | 0.75 | 0.96 | 0.98 |
| CLIP4CIR-H14-LAION2B | 82.5 | 0.84 | 0.84 | 0.85 | 0.94 | 0.63 | 0.98 | 0.68 | 0.87 | 0.80 | 0.95 | 0.98 | 0.39 | 0.81 | 0.97 | 0.98 |
| BLIP2-CIR | 63.2 | 1.01 | 0.99 | 1.01 | 1.01 | 0.76 | 1.00 | 0.75 | 0.90 | 0.88 | 0.96 | 0.97 | 0.46 | 0.84 | 1.00 | 0.99 |

| | | Noise | | | Blur | | | | Weather | | | | Digital | | | |
|---|---|---|---|---|---|---|---|---|---|---|---|---|---|---|---|---|
| **FashionIQ-C** | Clean | Gauss. | Shot | Implu. | Defoc. | Glass | Motion | Zoom | Snow | Frost | Fog | Bright | Contr. | Elast. | Pixel | JPEG |
| Pic2word | 24.7 | 0.61 | 0.61 | 0.59 | 0.57 | 0.38 | 0.79 | 0.61 | 0.54 | 0.54 | 0.66 | 0.73 | 0.38 | 0.43 | 0.85 | 0.82 |
| SEARLE | 25.6 | 0.64 | 0.64 | 0.63 | 0.62 | 0.37 | 0.85 | 0.63 | 0.56 | 0.56 | 0.69 | 0.78 | 0.39 | 0.44 | 0.87 | 0.92 |
| TIRG | 23.8 | 0.28 | 0.26 | 0.23 | 0.34 | 0.22 | 0.61 | 0.57 | 0.32 | 0.27 | 0.37 | 0.61 | 0.12 | 0.64 | 0.85 | 0.85 |
| MAAF | 23.4 | 0.31 | 0.27 | 0.25 | 0.44 | 0.21 | 0.67 | 0.53 | 0.29 | 0.24 | 0.31 | 0.54 | 0.13 | 0.54 | 0.83 | 0.83 |
| ARTEMIS | 24.9 | 0.24 | 0.24 | 0.20 | 0.38 | 0.26 | 0.65 | 0.60 | 0.36 | 0.25 | 0.38 | 0.55 | 0.14 | 0.63 | 0.86 | 0.87 |
| FashionViL | 23.4 | 0.26 | 0.28 | 0.25 | 0.40 | **0.31** | **0.82** | **0.67** | 0.33 | 0.31 | 0.34 | **0.70** | 0.15 | **0.86** | **1.09** | **1.06** |
| CLIP4CIR | **35.9** | **0.44** | **0.42** | **0.44** | **0.54** | 0.21 | 0.72 | 0.50 | **0.46** | **0.43** | **0.60** | **0.70** | **0.22** | 0.37 | 0.74 | 0.83 |

| | | Noise | | | Blur | | | | Weather | | | | Digital | | | |
|---|---|---|---|---|---|---|---|---|---|---|---|---|---|---|---|---|
| **CIRCO-C** | Clean | Gauss. | Shot | Implu. | Defoc. | Glass | Motion | Zoom | Snow | Frost | Fog | Bright | Contr. | Elast. | Pixel | JPEG |
| Image-only (CLIP) | 2.79 | 0.30 | 0.32 | 0.38 | 0.32 | 0.04 | 0.53 | 0.10 | 0.22 | 0.18 | 0.56 | 0.80 | 0.01 | 0.05 | 0.63 | 0.62 |
| Text-only (CLIP) | 2.51 | 0.60 | 0.54 | 0.59 | 0.53 | 0.08 | 0.88 | 0.45 | 0.45 | 0.41 | 0.6 | 0.79 | 0.06 | 0.11 | 0.84 | 0.80 |
| Pic2word | 8.50 | 0.65 | 0.70 | 0.67 | 0.78 | 0.19 | 0.90 | 0.42 | 0.60 | 0.38 | 0.65 | 0.83 | 0.21 | 0.29 | 0.84 | 0.77 |
| SEARLE | 15.1 | 0.64 | 0.69 | 0.64 | 0.70 | 0.18 | 0.86 | 0.34 | 0.52 | 0.37 | 0.61 | 0.81 | 0.10 | 0.30 | 0.86 | 0.81 |

Figure 3: Models average performance in CIRR-C under 15 vision corruptions. Left: Recall vs. rank K. Right: $\gamma$ vs. recall@10.

**Pretraining.** Among the compared models, FashionViL, CIRPLANT and CLIP-based methods (CLIP4CIR, Pic2word and SEARLE) are pretrained on large datasets, with respective sizes of 1.35 million, 6.5 million, and 400 million image-text pairs, while another three compared models are based on ImageNet pretrained

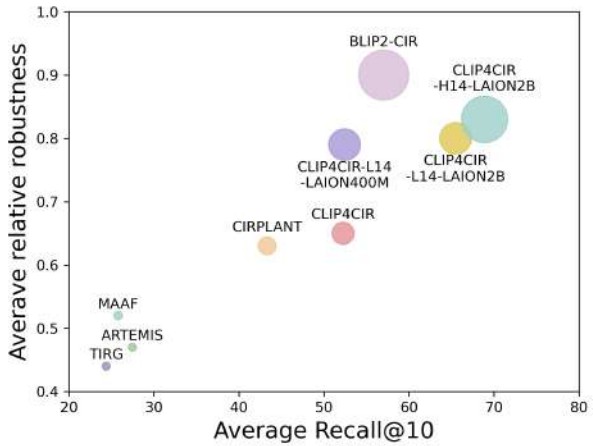 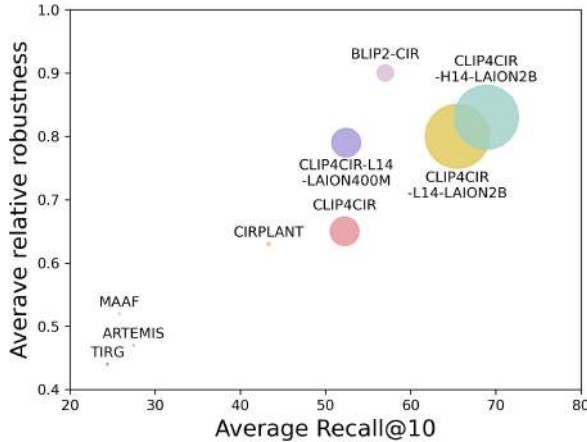

(a) Circle size indicates the number of model parameters.    (b) Circle size indicates the size of the pretrained dataset.

Figure 4: The average relative robustness γ versus the average Recall@10 of models under 15 vision corruptions in CIRR.

Table 3: Relative robustness score for text-image composed retrieval under 7 natural text corruptions in CIRR-C recall@10 and FashionIQ-C recall@10 on average of three categories. Recall@10 performance under clean conditions on the left. **Bold** are the highest relative robustness among the compared methods.

| | | | Character | | | | Word | |
| --- | --- | --- | --- | --- | --- | --- | --- | --- |
| **CIRR-C** | Clean | Swap | QWERTY | RemoveC. | RemoveS. | Misspelling | Repetition | Homophone |
| Text-only(CLIP) | 51.2 | 0.75 | 0.74 | 0.78 | 1.0 | 0.99 | 0.98 | 0.92 |
| Pic2word Saito et al. (2023) | 64.8 | 0.87 | 0.85 | 0.87 | 1.0 | 0.99 | 0.99 | 0.94 |
| SEARLE Baldrati et al. (2023) | 67.2 | 0.89 | 0.88 | 0.89 | 1.0 | 1.0 | 1.0 | 0.96 |
| TIRG Vo et al. (2019) | 55.1 | 0.77 | 0.76 | 0.80 | 1.0 | 0.98 | 1.0 | 0.89 |
| MAAF Dodds et al. (2020) | 49.9 | **0.95** | **0.97** | **0.96** | **1.0** | **1.0** | **1.0** | **0.97** |
| ARTEMIS Delmas et al. (2022) | 59.0 | 0.61 | 0.58 | 0.65 | **1.0** | 0.98 | 0.98 | 0.82 |
| CIRPLANT Liu et al. (2021) | 68.8 | 0.92 | 0.93 | 0.93 | **1.0** | **1.0** | **1.0** | **0.97** |
| CLIP4CIR Baldrati et al. (2022) | **80.3** | 0.89 | 0.89 | 0.90 | **1.0** | **1.0** | 0.99 | **0.97** |

| | | | Character | | | | Word | |
| --- | --- | --- | --- | --- | --- | --- | --- | --- |
| **FashionIQ-C** | Clean | Swap | QWERTY | RemoveC. | RemoveS. | Misspelling | Repetition | Homophone |
| Pic2word Saito et al. (2023) | 24.7 | 0.51 | 0.51 | 0.56 | 0.69 | 0.69 | 0.68 | 0.64 |
| SEARLE Baldrati et al. (2023) | 29.1 | 0.53 | 0.54 | 0.57 | 0.70 | 0.69 | 0.67 | 0.65 |
| TIRG Vo et al. (2019) | 23.8 | 0.26 | 0.20 | 0.29 | 0.66 | 0.63 | 0.61 | 0.52 |
| MAAF Dodds et al. (2020) | 23.4 | 0.40 | 0.39 | 0.39 | 0.70 | 0.68 | 0.68 | 0.62 |
| ARTEMIS Delmas et al. (2022) | 24.9 | 0.25 | 0.20 | 0.31 | 0.70 | 0.67 | 0.67 | 0.55 |
| FashionViL Han et al. (2022b) | 23.4 | **0.55** | **0.59** | **0.60** | **0.86** | **0.84** | **0.85** | **0.76** |
| CLIP4CIR Baldrati et al. (2022) | **35.9** | 0.52 | 0.51 | 0.54 | 0.71 | 0.70 | 0.69 | 0.67 |

ResNet50 as image encoder and random initialized LSTM as text encoder. As shown in Figure 4, the models with large pretrained datasets consistently show better robustness in both the open domain and fashion domain. *This implies that models with large pretrained datasets may result in better robustness against visual corruptions, which is in alignment with the findings from Paul et al. Paul & Chen (2022).*

**Bottleneck of robustness.** In Figure 3, we present visualizations of the recall performance with rank K improvement on the left and relative robustness on the right. With ranking $K$ improves, both recall and relative robustness improve for all models. Additionally, the ResNet50-based image-only search shows superior accuracy as well as robustness compared to TIRG, ARTEMIS, and MAAF, which share the same image encoder, ResNet50. According to the three foundations of text-image composed retrieval in Sec. 3, both the

image encoder and modality fusion module can be vulnerable to corruption in the visual domain. We observe that the ResNet50 backbone shows relatively high robustness, while the modality fusion of TIRG, MAAF, and ARTEMIS models exacerbates the instability of the model. However, this observed phenomenon does not apply to the CLIP features, whose text and image embedding are aligned in a unified space in the pretraining process. Comparing Image-only (CLIP) and CLIP4CIR, which query with CLIP image embedding and CLIP text-image composed embedding respectively, we can find out CLIP4CIR consistently performs better recall performance as well as robustness in Figure 3. *Thus, we speculate that text features from a shared vision-language space can help improve robustness, while text features from independent spaces will damage the model robustness.*

Further to better pinpoint the vulnerability in various model fusion modules, we compare TIRG, MAAF and ARTEMIS with the same text LSTM and image backbone ResNet50 but different fusion methods. As shown in Table 2 in the open domain, MAAF shows the trend performing the most robust, while ARTEMIS performs the second best over TIRG. The modality fusion modules of these three models, TIRG, ARTEMIS, and MAAF are concatenation-based, light attention, and transformer, respectively. Among them, MAAF utilizes modality-agnostic attention which extracts word and image tokens to conduct thorough merge through self-attention and cross-attention. *We hypothesize that more sufficient cross-modal interactions, such as cross-attention, can better promote robustness.*

### 4.2  Textual robustness against natural corruption

Comparing the relative robustness against textual corruption in Table 3 and visual corruption in Table 2, we can observe that the robustness is higher against textual corruptions. Among the compared models, MAAF and FashionViL show the highest robustness in the open domain and fashion domain respectively. This aligns with our findings regarding corruptions to the visual domain, where large pretrained models (FashionViL with fashion-specific pretraining) and models with sufficient modality fusion result in higher model robustness. Additionally, comparing CLIP4CIR and the Text-only retrieval method implementing CLIP text embedding, we observe that robustness can be increased after fusion with the vision modality. However, with images corrupted, CLIP4CIR shows lower robustness than text-only models, from which *we speculate that aligned clean image features can boost the robustness, while the corrupted image features will impair robustness.*

### 4.3  Robustness against text understanding

In this section, we analyze model reasoning ability through variation of modified text on numerical variation, attributes variation, object removal, background variation, and fine-grained variation, which are provided by our proposed CIRR-D dataset. We evaluate the performance for each query type as shown in Table 4. To compare the model's understanding of different text inputs, we evaluate the performance on the CIRR-D gallery set across various queries involving variations in numerical, attribute, object removal, and background. To establish a baseline mixed of different query categories, we utilize the 4181 queries from the origin CIRR dataset and evaluate their performance in our extended CIRR-D dataset. This baseline includes diverse reasoning instructions and aims to represent the models' average performance across various types of instructions. The comparison between the baseline and the specific query types is regarded as the understanding ability of each specific type, involving variations in numerical aspects, attributes, object removal and background. However, unlike the above four categories, the gallery set for fine-grained is a subset, which is composed of six highly similar images following Liu et al. (2021). Detailed analyses are discussed below.

**Numerical variation.** To probe the ability of numerical variation, the modified text contains either a precise value of the number from zero to ten or an estimated value by comparison e.g. '*reduce/increase the number*'. Comparing numerical specific queries with CIRR queries as shown in Table 4, we do not observe significant variation which may result from the long-tailed distribution. Namely, the numerical set has a large number of samples in the range of 1 to 3, while a small number of samples in the range of 4 to 10. More analysis can be found in the supplementary D. For now, we speculate that *numerical modification may not be the bottleneck of the current text-image composed retrieval.*

Table 4: Recall of CIRR-D dataset. The red and green arrows indicate the performance increase or decrease compared with CIRR queries. **Bold** and underline are the largest decrease and increase.

| | | | | R@5 | | Rsub@1 |
|---|---|---|---|---|---|---|
| | CIRR | Numerical | Attribute | Removal | Background | Fine grained |
| Image-only(RN50) | 31.55 | 31.47 ↓ (0.08) | 32.57 ↑ (1.02) | 35.99 ↑ (4.44) | 39.15 ↑ (7.60) | 20.25 |
| Image-only(CLIP) | 22.51 | 24.80 ↑ (2.29) | 29.09 ↑ (6.58) | 27.90 ↑ (5.39) | 25.64 ↑ (3.13) | 20.02 |
| Text-only(CLIP) | 39.02 | 42.84 ↑ (3.82) | 49.45 ↑ (10.43) | 11.62 ↓ (27.4) | 11.62 ↓ (**27.4**) | 53.73 |
| TIRG Vo et al. (2019) | 36.35 | 39.64 ↑ (3.29) | 37.77 ↑ (1.42) | 30.41 ↓ (5.94) | 32.82 ↓ (3.53) | 35.90 |
| MAAF Dodds et al. (2020) | 32.19 | 32.53 ↑ (0.34) | 35.57 ↑ (3.38) | 31.09 ↓ (1.10) | 34.27 ↑ (2.08) | 28.63 |
| ARTEMIS Vo et al. (2019) | 40.05 | 39.56 ↓ (0.49) | 42.68 ↑ (2.63) | 33.26 ↓ (6.79) | 35.56 ↓ (4.49) | 40.80 |
| CIRPLANT Liu et al. (2021) | 48.82 | 45.07 ↓ (**3.75**) | 47.73 ↓ (**1.09**) | 41.12 ↓ (7.70) | 45.98 ↓ (2.84) | 38.19 |
| CLIP4CIR Baldrati et al. (2022) | 62.94 | 64.18 ↑ (1.24) | 69.15 ↑ (6.21) | 31.66 ↓ (**31.28**) | 41.88 ↓ (21.06) | 62.66 |

**Attributes variation.** To evaluate the model's discriminative ability when querying elementary attributes, the modified text includes variations of color, shape and size. As observed in Table 4, all of the methods (except CIRPLANT) achieve higher performance with attribute queries than with CIRR queries. Additionally, the performance of CLIP based image-only model and CLIP4CIR have an obvious increment of over 6% compared with their performance with CIRR queries, which yield strong performance for attribute recognition including color, shape, and size. This implies that *attributes are one of the main focus points during training and models gain strong attribute discriminative ability.*

**Object removal.** Object removal is a convenient way to describe the differences between images but is universally overlooked by current methods in text-image composed retrieval. To probe the ability of object removal through CIRR-D, the modified text of the query explicitly contains the word *'remove'*. As shown in Table 4, all of the five compared methods achieve their lowest performance in object removal with an average decrement of 10.6% compared with the CIRR query. In particular, CLIP4CIR has a performance drop of over 30%, which may be a result of its static pretraining process, aligning only image text pairs without comparison between images. Surprisingly, image-only methods can have an increment over CIRR queries, which illustrates that visual similarity can boost the robustness over object removal but the text condition over guidance the model decision. This aligns with the foundation of the task: images are dense and continuous while text is sparse and discrete. *In the case of object removal, text guidance expands the possibility of the targets, which distracts the model and results in lower performance.*

**Background variation.** To probe the robustness against background modifications, the modified text of the query explicitly includes the word "background". We observe a similar phenomenon as in object removal, where the performance of compared models (except MAAF) decreases but the performance of image-only models increases compared to CIRR queries. As shown in the CIRR-D sample visualization in Figure 2, we can observe that the background modification method is limited such as changing the background color or making the background blur, which can lead to unrelated targets by relying solely on the text itself. We further speculate that *a modified textual description leading to a larger number of satisfactory candidate images may result in worse results.*

**Fine-grained variation.** To probe the fine-grained variation discriminative ability, we utilize the subset in the CIRR dataset, where each image is retrieved from its subset composed of another five highly similar images. As the gallery is different from the above reasoning function, the recall cannot be compared with CIRR query performance directly. We can observe from Table 4 that image-only models perform similarly to random guessing and text-only models using CLIP embeddings can achieve an acceptable result. Among the five compared methods, TIRG achieves the lowest performance which suggests the slight adjustment in visual space is sufficient rather than exploring text deeply and establishing a fusion space. This phenomenon indicates that it is difficult to distinguish between two states in continuous visual space. In contrast, text can precisely define subtle differences due to its discrete nature. We also speculate that *a modified text offers accurate information while minimizing the number of feasible targets, enhancing the model's discriminative ability.*

## 5    Conclusion

In this work, we proposed three robustness benchmarks for text-image composed retrieval–including two for natural corruption (in both images and text) and one for probing textual understanding. Concretely, we first introduced two benchmark datasets, CIRR-C and FashionIQ-C, with natural corruption in the open and fashion domains respectively. Additionally, we created the benchmark dataset CIRR-D to assess textual understanding: including under variations to numerical object count, different attributes, object removal, background change, and fine-grained variation. Based on our results, we provide the following observations about enhancing model robustness in text-image composed retrieval: 1) Models pretrained on large datasets will lead to better robustness, 2) Text features from an aligned space can help boost the robustness, while text features from independent spaces will reduce robustness, and 3) A modification to the textual query is more likely to enhance the model's discriminative ability when it minimizes the number of feasible targets (and will harm performance when producing more candidate responses). We suggest that these findings have the potential to boost the robustness of text-image composed retrieval in future work.

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

# A    Creating Benchmark Datasets

We build three benchmark datasets for this work evaluating both natural corruption (CIRR-C and FashionIQ-C) and textual understanding (CIRR-D). To evaluate natural corruption with both image and text, we introduce CIRR-C and FashionIQ-C based on the existing dataset CIRR and FashionIQ. To evaluate textual understanding including variations of numerical, attributes (colour, shape and size), object removal, background and fine-grained details, we introduce CIRR-D by categorizing and expanding CIRR with synthetic images. We provide raw images and complete code for generating all types of natural corruptions and the evaluation testbed in our code zip file. For shortcut, we provide raw image link for CIRR, FashionIQ and CIRR-D. To implement **CIRR-D** dataset, both raw images and queries in different categories (numerical, attribute, object removal, background and fine-grained variations) are provided directly. To implement **CIRR-C** and **FashionIQ-C** dataset, research can recreate the same benchmark datasets with the following steps:

1. Download CIRR and FashionIQ raw images with our provided link.

2. Preprocess image or text with the provided code of image corruption and text corruption.

3. Apply the proposed corruptions with our testbed for downstream model evaluation.

# B    Sample visualization

### B.1    CIRR-C visualization

We show the visualization samples from CIRR-C in Figure. 8. Our CIRR-C is based on the CIRR dataset and implemented with both image corruptions and text corruptions. We apply 15 standard natural image corruptions, as depicted in Figure 8 (a), and demonstrate 5 levels of severity using brightness corruption as an example in Figure 8 (b). We further visualize 7 text corruptions in Figure 8 (c). For both image and text corruption, humans can easily recognize them.

### B.2    FashionIQ-C visualization

FashionIQ-C follows the same natural corruption in both image and text as in CIRR-C. We show the visualization samples from FashionIQ-C in Figure. 9. FashionIQ-C is based on the FashionIQ dataset and implemented with both image corruptions and text corruptions. We apply 15 standard natural image corruptions, as depicted in Figure 9 (a), and demonstrate 5 levels of severity using zoom blur corruption as an example in Figure 9 (b). We further visualize 7 text corruptions in Figure 9 (c).

### B.3    Textual corruption definition

In this work, we implement 7 natural textual corruptions following Rychalska et al. (2019). The definition of the textual corruptions are as follows:

- Swap: Randomly shuffles two characters within a word.

- Qwerty: Simulates errors made while writing on a QWERTY-type keyboard. Characters are swapped for their neighbors on the keyboard

- RemoveChar: Randomly removes characters from words.

- RemoveSpace: Removes a space from text, merging two words.

- Misspelling: Misspells words appearing in the Wikipedia list of commonly misspelled English words.

- Repetition: Randomly repeat words.

- Homophone: Changes words into their homophones from the Wikipedia list of common homophones. The list contains around 500 pairs or triples of homophonic words.

Table 5: Details of CIRR-D dataset. The first column is the number of images. The rest columns contain the number of triplets for five probing abilities.

|  | Images | Numerical | Attribute | Removal | Background | Fine-grained |
|---|---|---|---|---|---|---|
| Val. | 2297 | 820 | 1397 | 233 | 358 | 4181 |
| Extend caption | - | - | - | 505 | 812 | - |
| Synthetic | 1245 | 305 | 700 | 140 | - | - |
| Total | 3542 | 1125 | 2097 | 878 | 1170 | 4181 |

Examples are shown in Figure. 8 for CIRR-C and Figure. 9 for FashionIQ-C respectively.

### B.4 CIRR-D visualization

To detect textual understanding ability, we build a CIRR-D dataset with five different types of queries containing specific instructions to probe five different abilities The source of the CIRR-D dataset is from the original CIRR, CIRR extends caption and our generated synthetic images. The triplets from the original CIRR dataset are normally with obvious variations while the synthetic triplets are normally following the same structure and only local variations. Some extended caption from the original CIRR dataset can only supply partial difference and cannot locate the target images. Therefore, we manually remove samples and retain only those triplets where the extended caption can provide sufficient variations. In detail, we visualize numerical samples in Figure 10, which is composed of triplets from the original CIRR dataset in Figure 10 (a) and our generated synthetic triplets in Figure.10 (b). Attribute variation visualization samples are shown in Figure 11, which is composed of triplets from the original CIRR dataset in Figure 11 (a) and our generated synthetic triplets in Figure. 11 (b). To evaluate object removal ability, the triplet source consists of three aspects. We visualize object removal triplets from the original CIRR dataset in Figure. 12 (a), extended caption triplets in Figure. 12 (b) and our generated synthetic triplets in Figure. 12 (c). we visualize background variations samples in Figure 13, which is composed of triplets from original CIRR dataset in Figure 13 (a) and from extended captions in Figure 13 (b). The evaluation of fine-grained variations follows the original CIRR dataset, whose gallery set is composed of 5 highly similar images as shown in Figure. 14.

## C More experiments result

### C.1 Fine grained subset analysis of CIRR-C

In this section, we supplement some experimental results. As shown in Figure. 5, we visualize the recall performance on the CIRR-C subset. Comparing the subset retrieval with the whole gallery in CIRR-C (shown in the main paper Figure.3), we can observe that subset relative robustness (range from 0.6 to 0.9) overall is higher than the whole set (range from 0.4 to 0.8). This result suggests that a smaller gallery can lead to more stable retrieval. In essence, the overall trend aligns with retrieval on all images: CLIP4CIR consistently performs the best, while IMAGE-ONLY with CLIP embedding consistently exhibits the worst retrieval performance.

### C.2 Subcategories analysis of FashionIQ-C

For detailed results in the FashionIQ-C dataset, we report the results on the three categories, namely dress, shirt and toptee respectively, as shown in Table. 6. Overall, a similar trend is observed across the three categories, with FashionViL and CLIP4CIR consistently exhibiting the highest relative robustness. In the shirt category, overall robustness tends to be slightly higher than in the dress and toptee categories. We further report the recall@10 performance on FashionIQ-C dataset as shown in Table. 7. By comparing the relative robustness in Table. 6 and corresponding recall performance in Table. 7, we can find out higher robustness doesn't mean higher recall performance. As according to the definition of relative robustness:

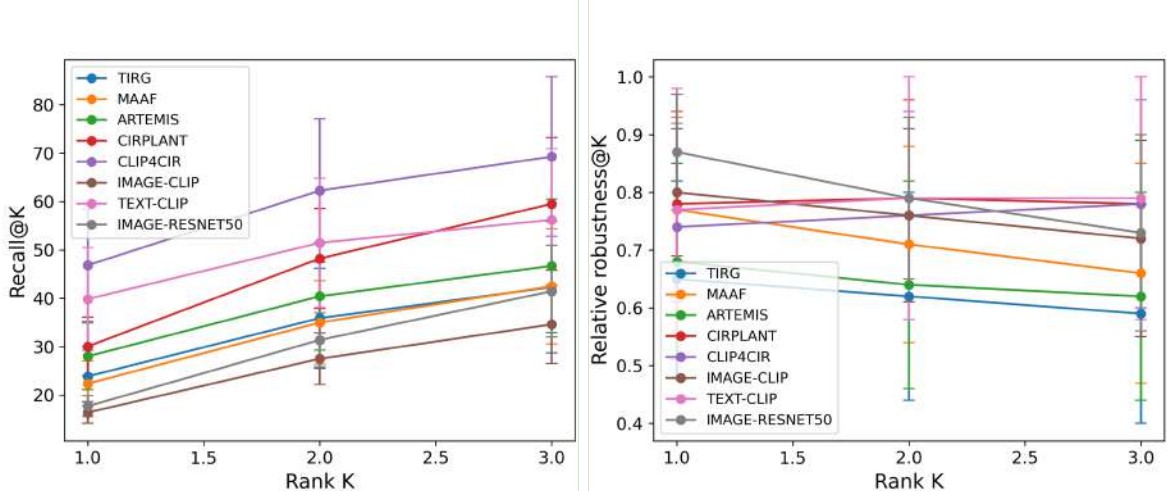

Figure 5: Models average performance in CIRR subset under 15 vision corruptions. Left: Recall vs. rank K. Right: Relative robusentess vs. rank K.

Table 6: Relative robustness score for text-image composed retrieval under 15 natural image corruptions in FashionIQ-C Recall@10 for dress, shirt and toptee respectively. **Bold** is the highest relative robustness for the five composed retrieval methods.

| | Noise | | | Blur | | | | Weather | | | | Digital | | | |
|---|---|---|---|---|---|---|---|---|---|---|---|---|---|---|---|
| **FashionIQ-C Dress** | Gauss. | Shot | Implu. | Defoc. | Glass | Motion | Zoom | Snow | Frost | Fog | Bright | Contr. | Elast. | Pixel | JPEG |
| TIRG Vo et al. (2019) | 0.21 | 0.18 | 0.17 | 0.37 | 0.22 | 0.64 | 0.58 | 0.33 | 0.25 | 0.35 | 0.63 | 0.12 | 0.62 | 0.82 | 0.85 |
| MAAF Dodds et al. (2020) | 0.30 | 0.24 | 0.22 | 0.42 | 0.19 | 0.65 | 0.56 | 0.28 | 0.21 | 0.32 | 0.58 | 0.10 | 0.54 | 0.78 | 0.81 |
| ARTEMIS Delmas et al. (2022) | 0.23 | 0.22 | 0.18 | 0.38 | 0.24 | 0.66 | 0.62 | 0.39 | 0.26 | 0.37 | 0.59 | 0.14 | 0.67 | 0.85 | 0.9 |
| FashionViL Han et al. (2022b) | 0.21 | 0.22 | 0.23 | 0.38 | **0.34** | **0.84** | **0.72** | 0.29 | 0.29 | 0.3 | **0.79** | 0.13 | **0.88** | **1.1** | **1.1** |
| CLIP4CIR Baldrati et al. (2022) | **0.44** | **0.38** | **0.44** | **0.54** | 0.24 | 0.74 | 0.52 | **0.41** | **0.36** | **0.55** | 0.68 | **0.16** | 0.42 | 0.75 | 0.82 |

| | Noise | | | Blur | | | | Weather | | | | Digital | | | |
|---|---|---|---|---|---|---|---|---|---|---|---|---|---|---|---|
| **FashionIQ-C Shirt** | Gauss. | Shot | Implu. | Defoc. | Glass | Motion | Zoom | Snow | Frost | Fog | Bright | Contr. | Elast. | Pixel | JPEG |
| TIRG Vo et al. (2019) | 0.33 | 0.32 | 0.27 | 0.28 | 0.20 | 0.57 | 0.54 | 0.32 | 0.28 | 0.37 | 0.51 | 0.15 | 0.60 | 0.86 | 0.81 |
| MAAF Dodds et al. (2020) | 0.33 | 0.30 | 0.27 | 0.46 | 0.20 | 0.67 | 0.50 | 0.30 | 0.27 | 0.34 | 0.47 | 0.16 | 0.57 | 0.84 | 0.79 |
| ARTEMIS Delmas et al. (2022) | 0.27 | 0.28 | 0.25 | 0.39 | **0.26** | 0.62 | **0.61** | 0.36 | 0.24 | 0.38 | 0.54 | 0.16 | 0.61 | 0.84 | 0.88 |
| FashionViL Han et al. (2022b) | 0.29 | 0.34 | 0.26 | 0.38 | **0.26** | **0.77** | 0.6 | 0.33 | 0.32 | 0.37 | 0.63 | 0.17 | **0.83** | **1.09** | **1.02** |
| CLIP4CIR Baldrati et al. (2022) | **0.47** | **0.48** | **0.45** | **0.50** | 0.18 | 0.65 | 0.48 | **0.51** | **0.50** | **0.65** | 0.71 | **0.27** | 0.31 | 0.69 | 0.82 |

| | Noise | | | Blur | | | | Weather | | | | Digital | | | |
|---|---|---|---|---|---|---|---|---|---|---|---|---|---|---|---|
| **FashionIQ-C Toptee** | Gauss. | Shot | Implu. | Defoc. | Glass | Motion | Zoom | Snow | Frost | Fog | Bright | Contr. | Elast. | Pixel | JPEG |
| TIRG Vo et al. (2019) | 0.30 | 0.28 | 0.25 | 0.36 | 0.24 | 0.63 | 0.58 | 0.32 | 0.27 | 0.39 | 0.58 | 0.10 | 0.69 | 0.88 | 0.88 |
| MAAF Dodds et al. (2020) | 0.30 | 0.28 | 0.27 | 0.45 | 0.24 | 0.71 | 0.52 | 0.28 | 0.23 | 0.28 | 0.56 | 0.14 | 0.52 | 0.88 | 0.88 |
| ARTEMIS Delmas et al. (2022) | 0.21 | 0.23 | 0.18 | 0.37 | 0.28 | 0.68 | 0.57 | 0.33 | 0.25 | 0.38 | 0.53 | 0.13 | 0.66 | 0.88 | 0.82 |
| FashionViL Han et al. (2022b) | 0.28 | 0.28 | 0.27 | 0.44 | **0.32** | **0.85** | **0.69** | 0.38 | 0.33 | 0.36 | 0.69 | 0.15 | **0.88** | **1.09** | **1.06** |
| CLIP4CIR Baldrati et al. (2022) | **0.42** | **0.4** | **0.42** | **0.58** | 0.21 | 0.76 | 0.49 | **0.46** | **0.44** | **0.60** | 0.71 | **0.24** | 0.39 | 0.78 | 0.84 |

Table 7: Recall@10 for text-image composed retrieval under 15 natural image corruptions in FashionIQ-C.

| | | Noise | | | Blur | | | | Weather | | | | Digital | | | |
|---|---|---|---|---|---|---|---|---|---|---|---|---|---|---|---|---|
| **FashionIQ-C** | Clean | Gauss. | Shot | Implu. | Defoc. | Glass | Motion | Zoom | Snow | Frost | Fog | Bright | Contr. | Elast. | Pixel | JPEG |
| TIRG Vo et al. (2019) | 23.8 | 6.6 | 6.1 | 5.4 | 8.1 | 5.3 | 14.6 | 13.5 | 7.7 | 6.3 | 8.8 | 13.8 | 3.0 | 15.2 | 20.3 | 20.2 |
| MAAF Dodds et al. (2020) | 23.4 | 7.2 | 6.4 | 5.9 | 10.4 | 5.0 | 15.8 | 12.3 | 6.6 | 5.5 | 7.3 | 12.7 | 3.1 | 12.7 | 19.4 | 19.4 |
| ARTEMIS Delmas et al. (2022) | 24.9 | 5.8 | 6.0 | 4.9 | 9.4 | 6.5 | 16.4 | 14.9 | 9.0 | 6.2 | 9.4 | 13.9 | 3.5 | 16.2 | 21.4 | 21.5 |
| FashionViL Han et al. (2022b) | 23.4 | 6.1 | 6.5 | 5.9 | 9.5 | 7.2 | 19.3 | 15.8 | 7.8 | 7.3 | 8.0 | 16.5 | 3.5 | 20.3 | 25.7 | 24.9 |
| CLIP4CIR Baldrati et al. (2022) | 35.9 | 15.9 | 15.2 | 15.6 | 19.4 | 7.5 | 25.7 | 17.8 | 16.5 | 15.6 | 21.5 | 25.2 | 8.2 | 13.3 | 26.5 | 29.7 |

$\gamma = 1 - (R_c - R_p)/R_c$ following Hendrycks & Dietterich (2019), lower recall performance under clean condition $R_c$ will lead to higher relative robustness $\gamma$.

Table 8: Relative robustness score for text-image composed retrieval under 15 natural image corruptions in COCO-C Recall@10.

| | Noise | | | Blur | | | | Weather | | | | Digital | | | |
|---|---|---|---|---|---|---|---|---|---|---|---|---|---|---|---|
| **COCO-C** | Gauss. | Shot | Implu. | Defoc. | Glass | Motion | Zoom | Snow | Frost | Fog | Bright | Contr. | Elast. | Pixel | JPEG |
| TIRG Vo et al. (2019) | 0.19 | 0.21 | 0.14 | 0.42 | 0.25 | 0.62 | 0.58 | 0.35 | 0.21 | 0.51 | 0.89 | 0.05 | 0.40 | 0.40 | 0.72 |
| ARTEMIS Delmas et al. (2022) | 0.14 | 0.16 | 0.08 | 0.43 | 0.22 | 0.72 | 0.52 | 0.41 | 0.32 | 0.45 | 1.06 | 0.05 | 0.40 | 0.48 | 0.70 |
| CLIP4CIR Baldrati et al. (2022) | 0.52 | 0.58 | 0.52 | 0.65 | 0.12 | 0.85 | 0.36 | 0.51 | 0.49 | 0.71 | 0.90 | 0.10 | 0.24 | 0.77 | 0.77 |

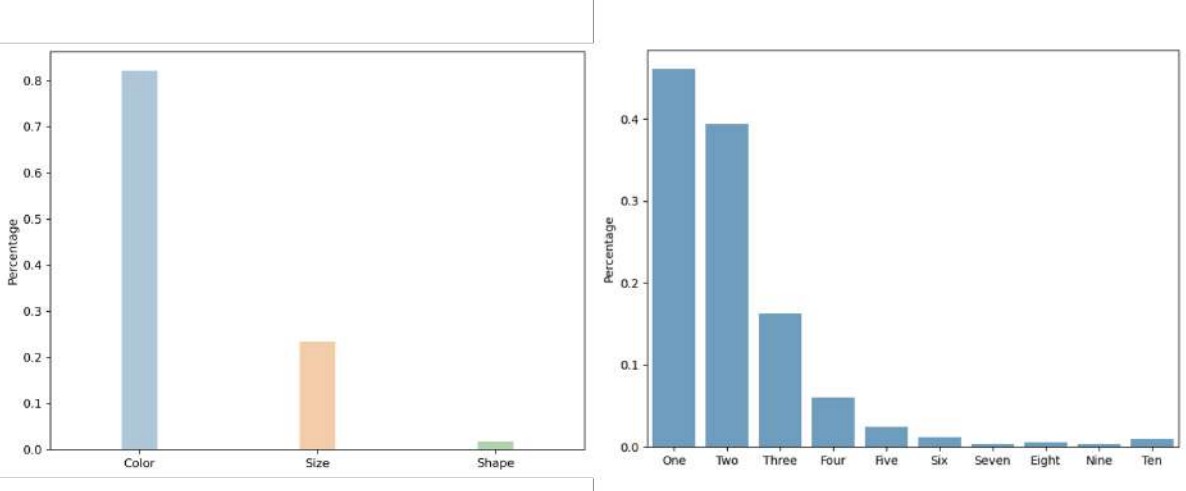

Figure 6: CIRR-D distribution for attribute variants and numerical variations.

## C.3 Analysis of COCO with image corruptions

To evaluate the compared models on more general domain, we implement our image corruptions on the validation set of COCO Lin et al. (2014), represented by CIRR-C. We set masked bounding box as the reference image, the raw image as the target image, and the labels of objects as modified text the following Neculai et al. (2022); Saito et al. (2023). The three compared models are trained on the CIRR dataset and evaluated on the validation set of COCO with 5000 images. The results show that large pretrained model CLIP4CIR have higher robustness than smaller models TIRG and ARTEMIS, which follow the same conclusion in paper Section 4.1.

## D  Limitation

We discuss the limitation of the proposed benchmarks in this section. For benchmarking natural corruption in CIRR-C and FashionIQ-C, the method of simulating real-world corruption with the noise still has limitations. For benchmarking textual understanding in CIRR-D, it has long-tail distribution. As shown in Figure. 6, both numerical and attribute evaluation set follows the long-tail distribution. The numerical set has a large number of samples in the range of 1 to 3, while each category from 4 to 10 has only a small number of samples. The attribute evaluation set has a large number of samples with colour variations and a small number of samples with size variations. The imbalanced distribution can lead to bias towards the categories with more data. Further, we visualise the performance of the query with number one to three, four to ten respectively shown in Figure 7 left. The average recall@5 of five evaluated methods are 43.06% on number one to three, 42.36% on numbers four to ten and 44.2% on number one to ten respectively. (A sentence with multiple numbers will be categorized to multiple categories, thus number one to three and number four to ten can overlap.) Based on this subtle accuracy change, we speculate that the model also possesses a similar capability for recognizing the less frequent samples (number four to ten) in the long-tail distribution as it does for the more frequent samples (number one to three).

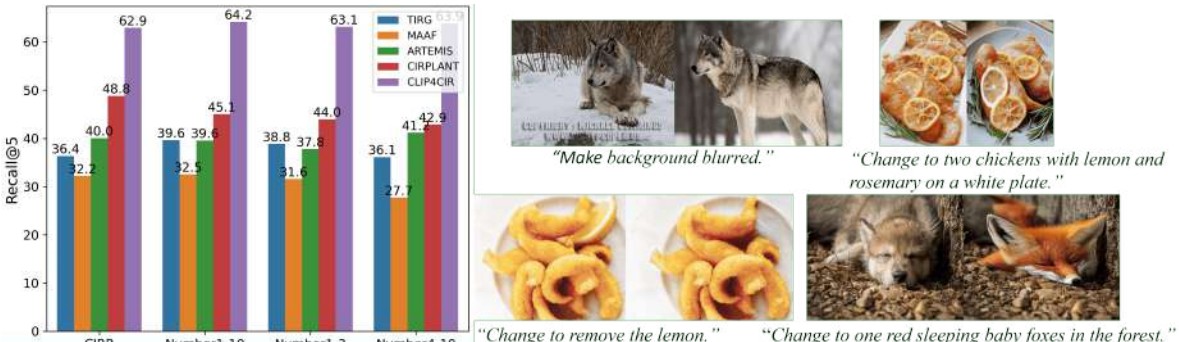

Figure 7: Left: Recall@5 on CIRR-D numerical queries. Right: Samples visualization of CIRR-D. The left and right images are the reference and target images. Except the upper left triplet, rest target images are synthetic.

## E    License

All the models in this study are available to the public. The model code for TIRG Vo et al. (2019) and MAAF Dodds et al. (2020) have the Apache License Version 2.0, ARTEMIS Delmas et al. (2022) has CC BY-NC-SA 4.0 License, CIRPLANT Liu et al. (2021) has MIT license and FashionViL Han et al. (2022b) has BSD License. We will provide CIRR-C, FashionIQ-C and CIRR-D publicly. These datasets are based on existing CIRR Liu et al. (2021) and FashionIQ Wu et al. (2021). For CIRR-C and FashionIQ-C, we didn't add any new images or text sources. For CIRR-D, we further generate synthetic images and text to expand the original CIRR dataset. All of these datasets are available to the public and we apply similar licenses to our testbed code and our proposed benchmarks.

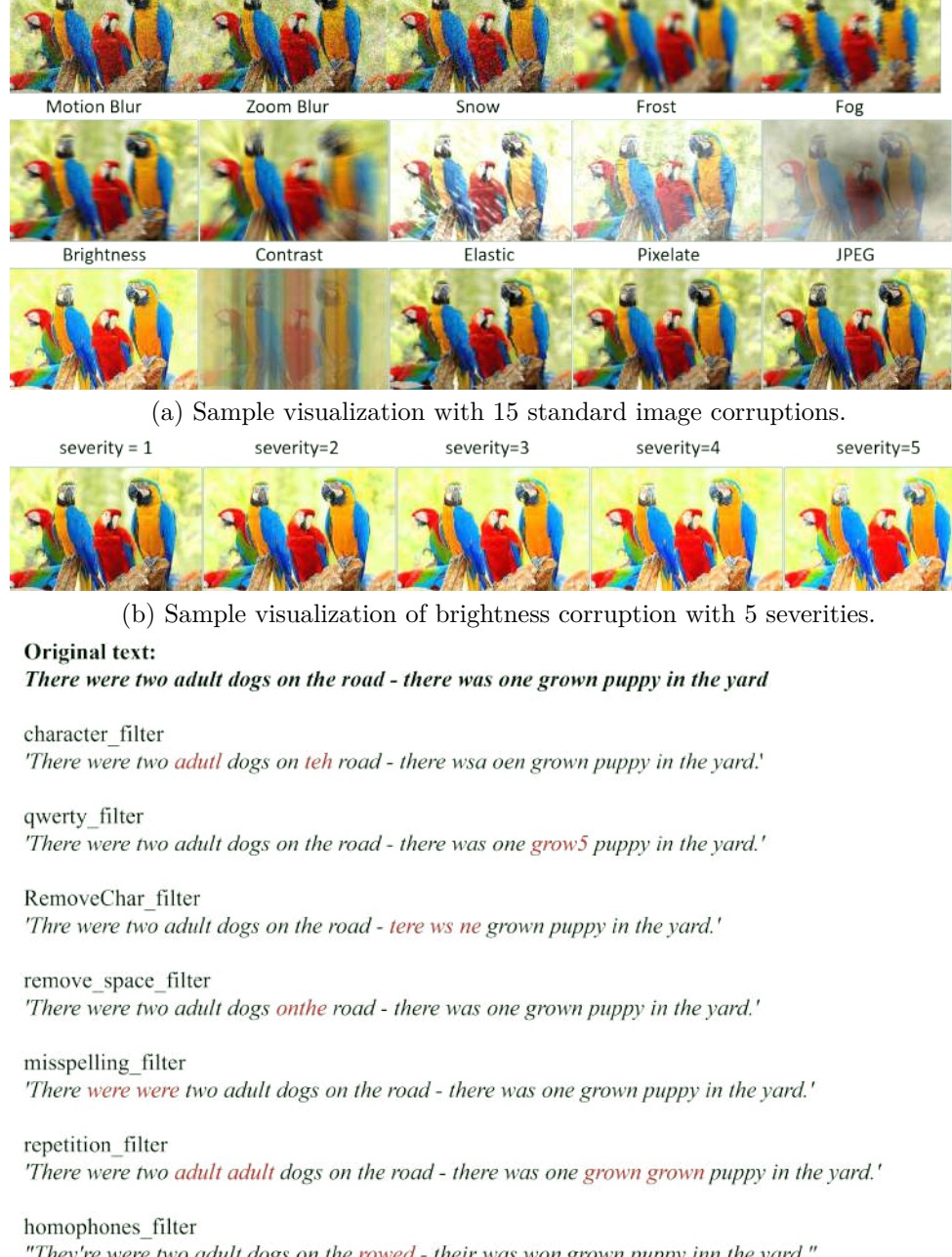

(a) Sample visualization with 15 standard image corruptions.

(b) Sample visualization of brightness corruption with 5 severities.

**Original text:**
*There were two adult dogs on the road - there was one grown puppy in the yard*

character_filter
*'There were two adutl dogs on teh road - there wsa oen grown puppy in the yard.'*

qwerty_filter
*'There were two adult dogs on the road - there was one grow5 puppy in the yard.'*

RemoveChar_filter
*'Thre were two adult dogs on the road - tere ws ne grown puppy in the yard.'*

remove_space_filter
*'There were two adult dogs onthe road - there was one grown puppy in the yard.'*

misspelling_filter
*'There were were two adult dogs on the road - there was one grown puppy in the yard.'*

repetition_filter
*'There were two adult adult dogs on the road - there was one grown grown puppy in the yard.'*

homophones_filter
*"They're were two adult dogs on the rowed - their was won grown puppy inn the yard."*

(c) Sample visualization of 7 text corruptions.

Figure 8: CIRR-C sample visualization: (a) 15 standard image corruptions, (b) 5 severities of brightness corruption and (c) 7 text corruption.

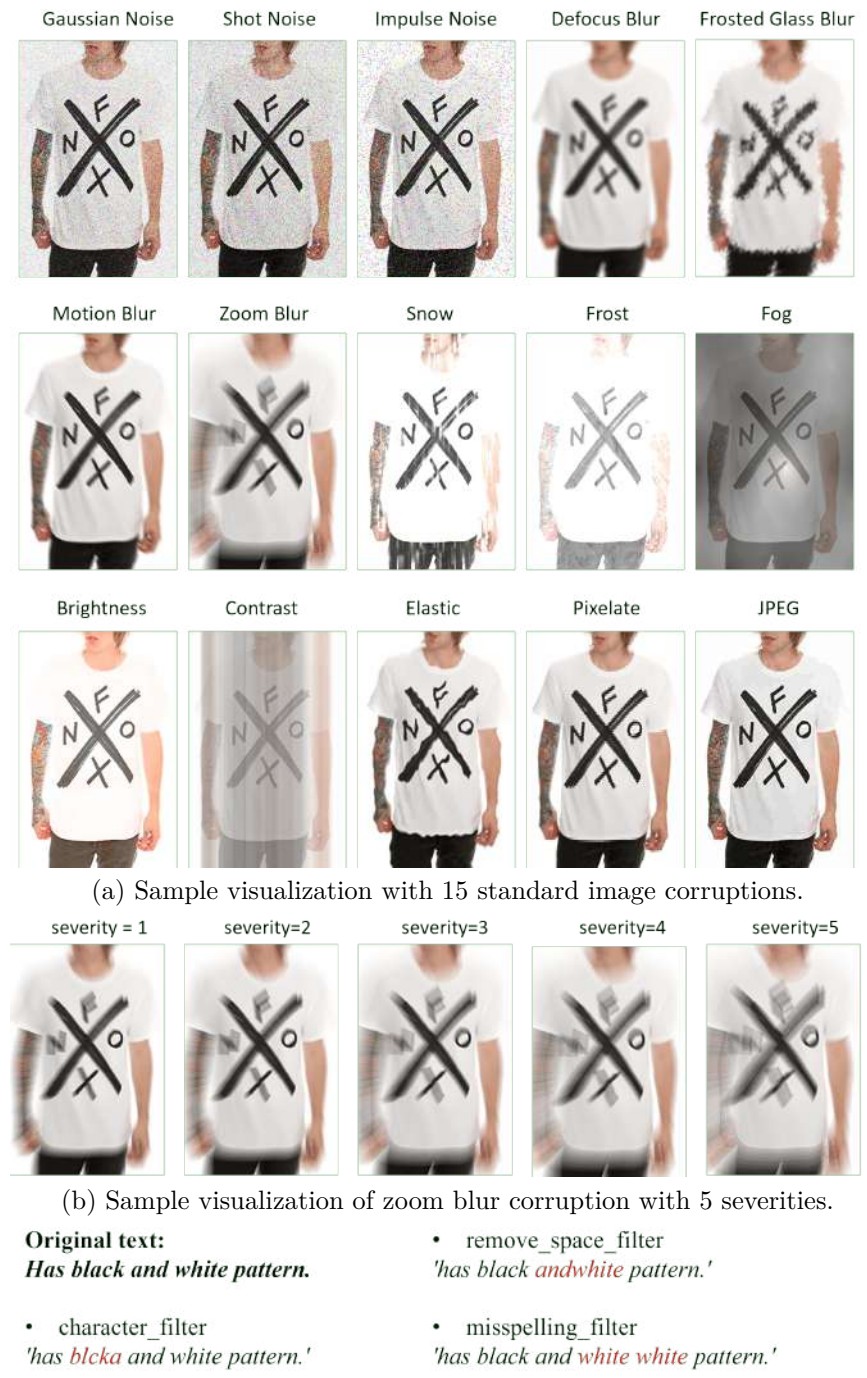

(a) Sample visualization with 15 standard image corruptions.

(b) Sample visualization of zoom blur corruption with 5 severities.

**Original text:**
**Has black and white pattern.**

- remove_space_filter
'has black *andwhite* pattern.'

- character_filter
'has *blcka* and white pattern.'

- misspelling_filter
'has black and *white white* pattern.'

- qwerty_filter
'has black and white *lattern.*'

- repetition_filter
'has *black black black* and white pattern.'

- RemoveChar_filter
'has black *ad* white *attern.*'

- homophones_filter
'has black and *wight* pattern.'

(c) Sample visualization of 7 text corruptions.

Figure 9: fashionIQ-C sample visualization: (a) 15 standard image corruptions, (b) 5 severities of zoom blur corruption and (c) 7 text corruptions.

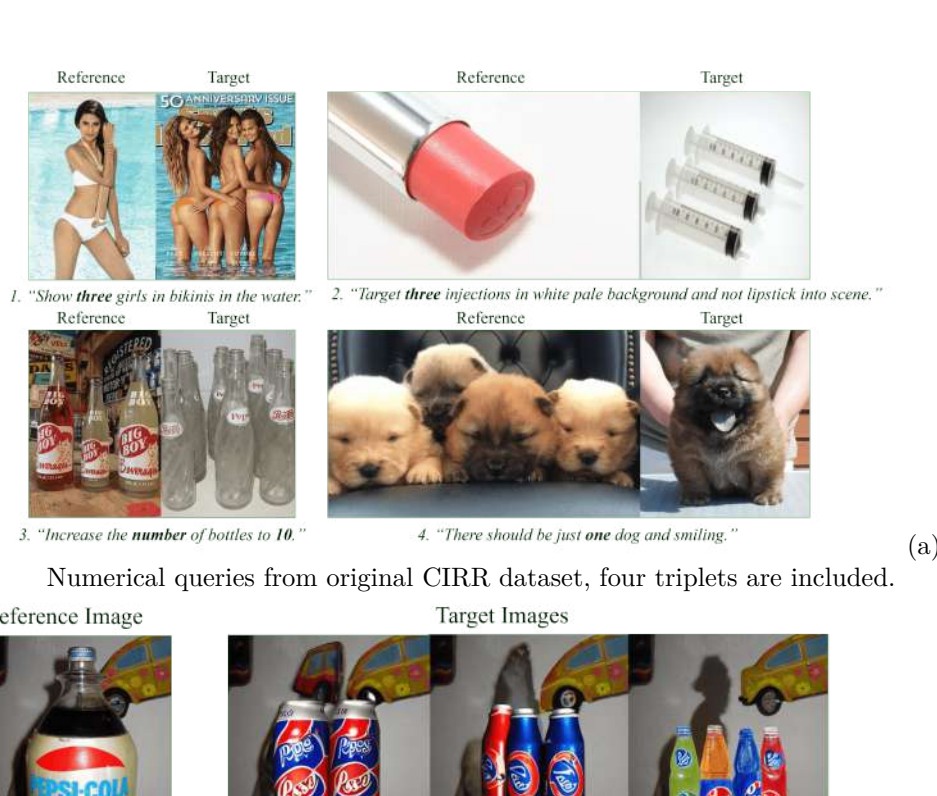

(a)
Numerical queries from original CIRR dataset, four triplets are included.

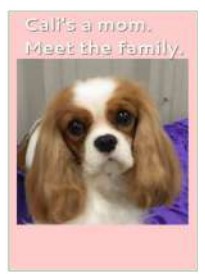
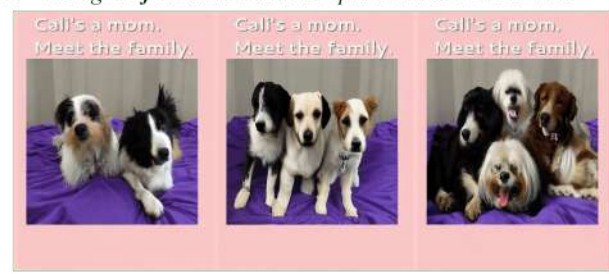

1. "Change to **two** blue and red Pepsi Colas are on the bus."
2. "Change to **three** blue and red Pepsi Colas are on the bus."
3. "Change to **four** blue and red Pepsi Colas are on the bus."

4. "Change to **two** dogs with pink backgrounds are lounging on the couch."
5. "Change to **three** dogs with pink backgrounds are lounging on the couch."
6. "Change to **four** dogs with pink backgrounds are lounging on the couch."

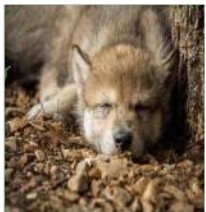
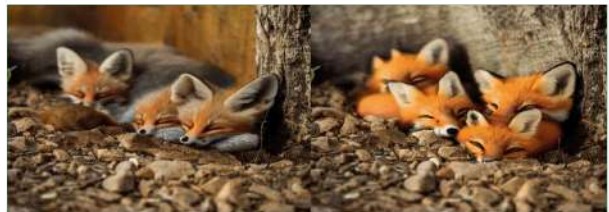

7. "Change to **three** sleeping baby foxes in the forest."
8. "Change to **four** sleeping baby foxes in the forest."

(b) Our generated numerical queries, eight triplets are included.

Figure 10: CIRR-D sample visualization for numerical queries.

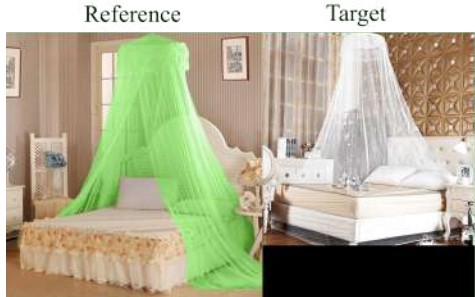

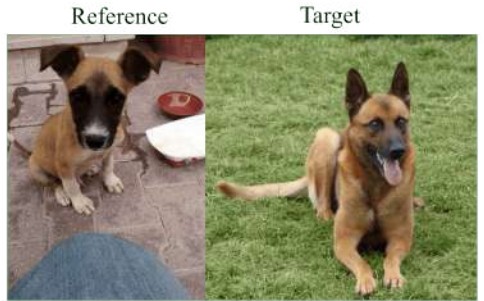

1. "Change the netting colour to white and make the background wall brown."

2. "Have a larger dog with pointier ears laying in the grass with its tongue out."

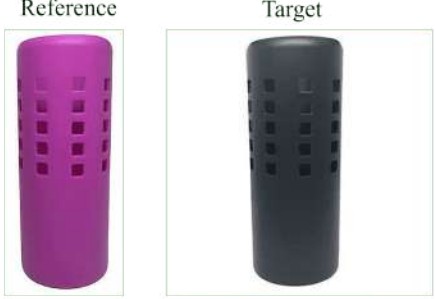

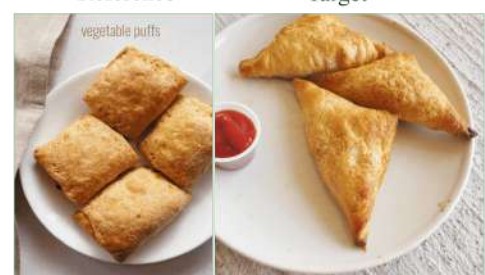

3. "Paint the tube black with square holes."

4. "Change the food into triangles and add a red sauce next to them."

(a) Attribute queries from original CIRR dataset, four triplets are included.

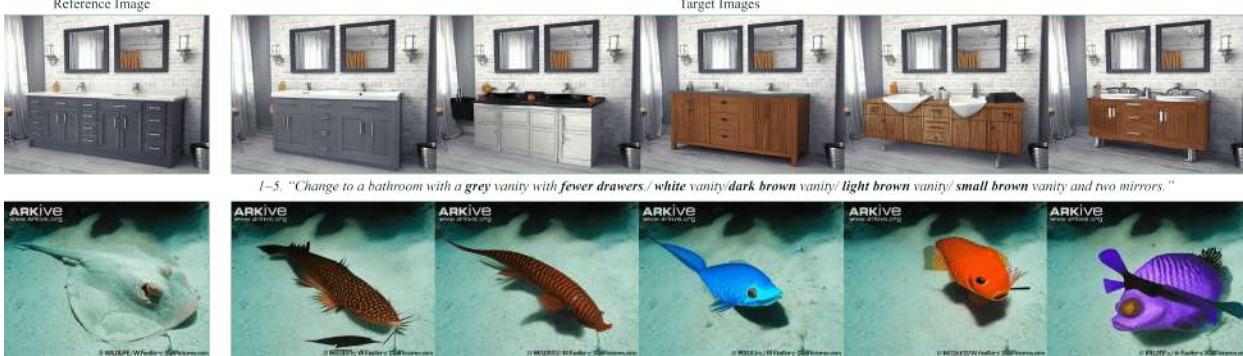

1~5. "Change to a bathroom with a grey vanity with fewer drawers./ white vanity/dark brown vanity/ light brown vanity/ small brown vanity and two mirrors."

6~10. "Change to a spotted red and black / striped / blue / an orange and yellow / purple stingfish in the sand."

(b) Our generated attribute queries, 10 triplets are included.

Figure 11: CIRR-D sample visualization for attribute queries including color, shape and size.

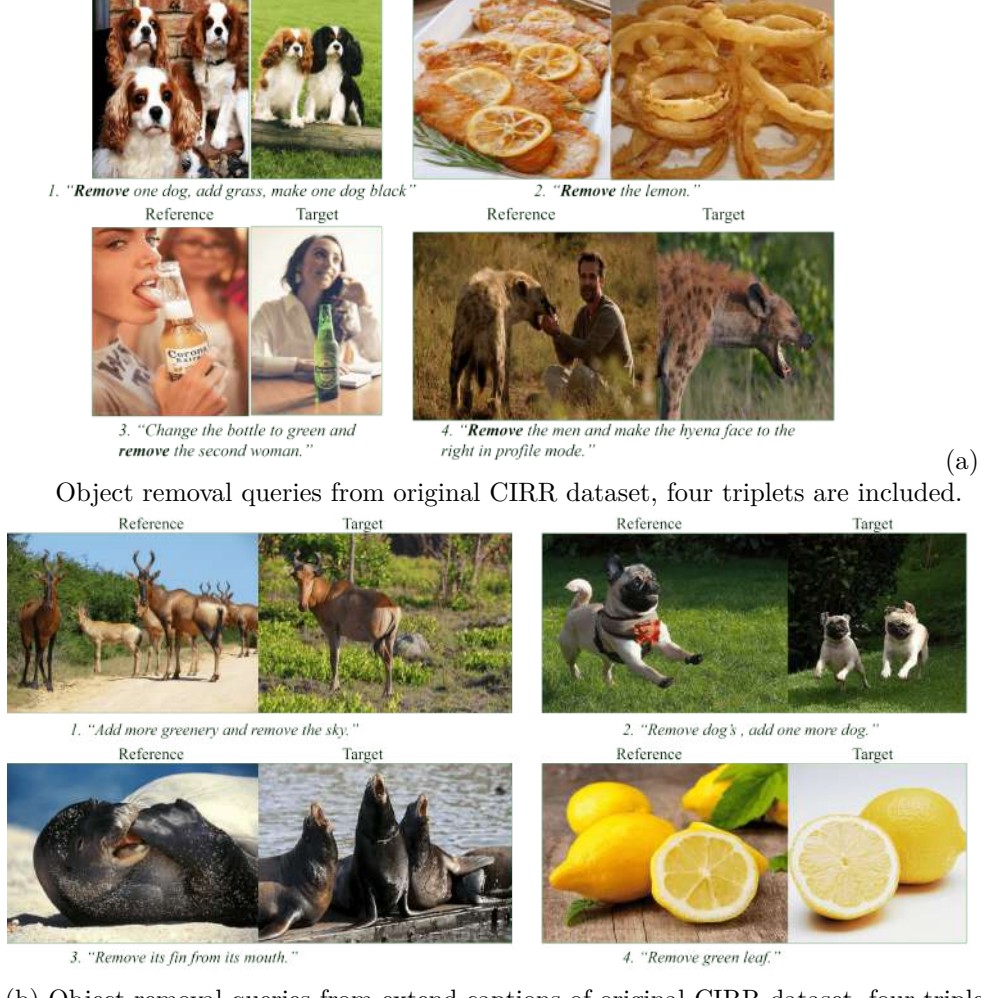

(a)

Object removal queries from original CIRR dataset, four triplets are included.

(b) Object removal queries from extend captions of original CIRR dataset, four triplets are included.

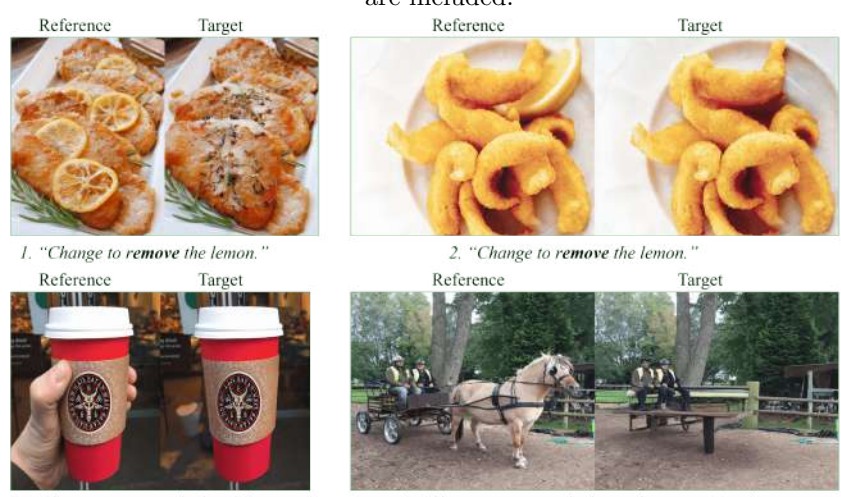

(c) Our generated object removal queries, four triplets are included.

Figure 12: CIRR-D sample visualization for object removal queries.

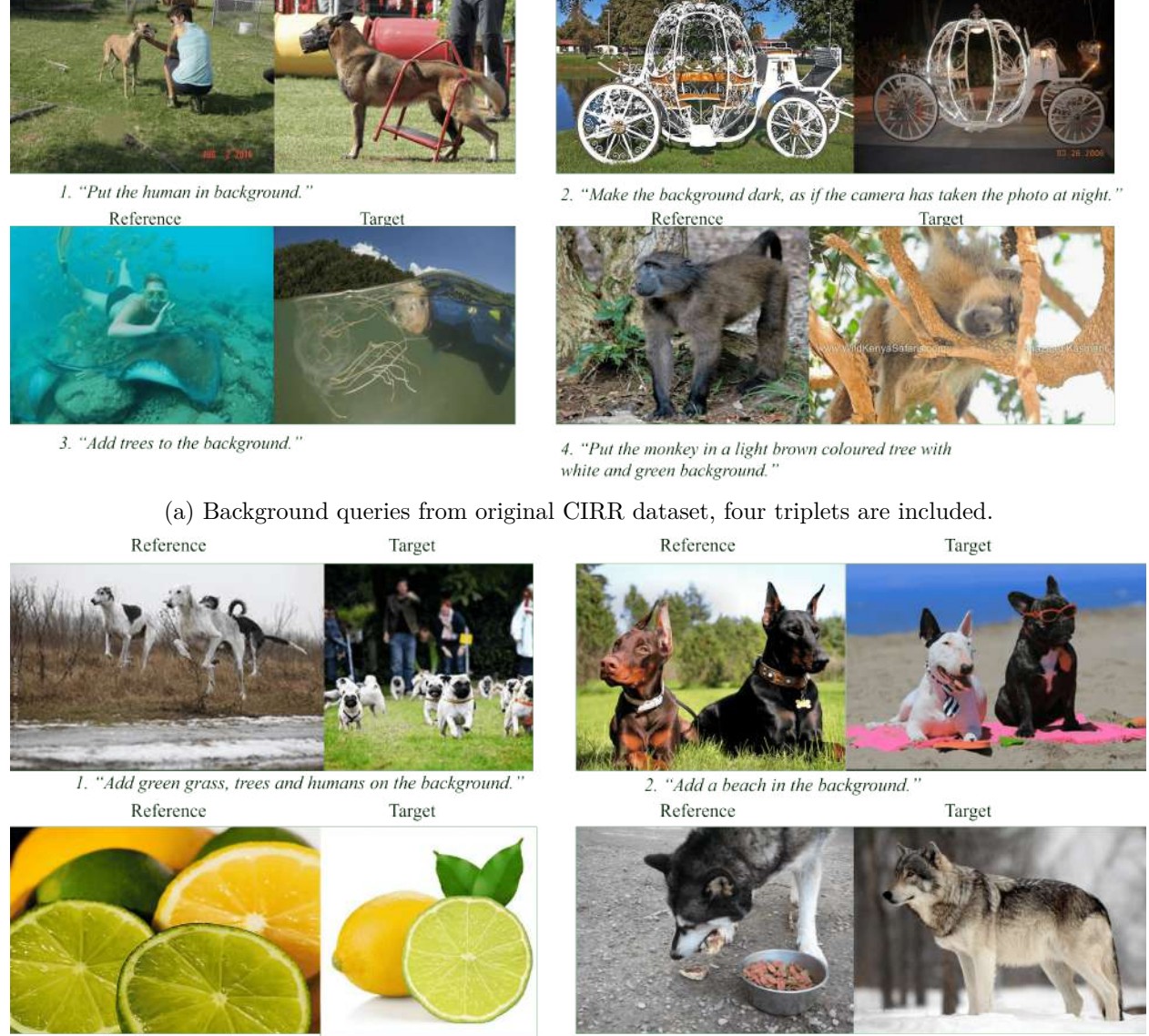

(a) Background queries from original CIRR dataset, four triplets are included.

(b) Background variation queries from extend captions of original CIRR dataset, four triplets are included.

Figure 13: CIRR-D sample visualization for background variations.

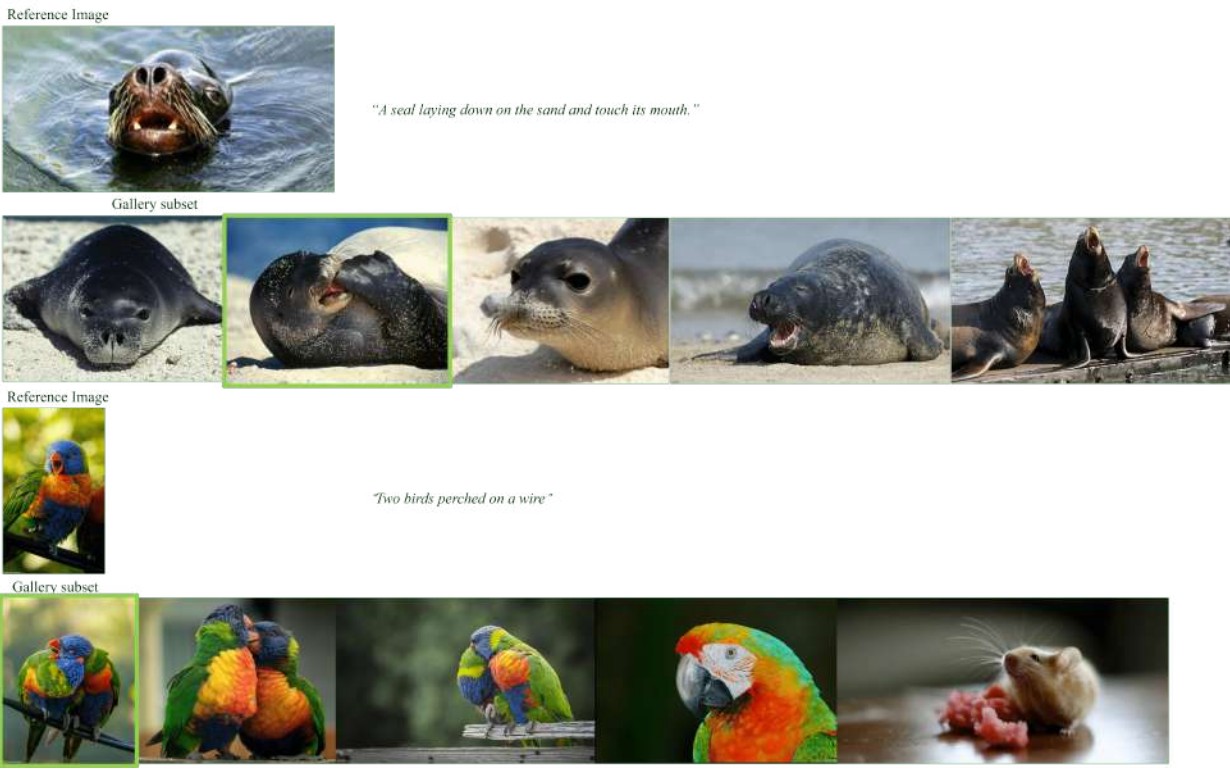

Figure 14: CIRR-D sample visualization for fine-grained variation queries, 2 triplets are included. The images with a green border are the correct targets, while the other images are highly similar composing the gallery set.

