# OpenReview forum: "Benchmarking Robustness of Text-Image Composed Retrieval"
_TMLR — Rejected by TMLR_

### Review · Reviewer_6xaX · 2023-12-13

**Summary Of Contributions:**

This paper provides a set of benchmarks that evaluate the robustness of models that aim, based on a template image and a text modifier, to retrieve images from a database that match the template and desired modifications from the text (image:'picture of a cat' + text: '2 cats' -> image:'picture of two cats'). This is denoted as Text-Image Composed Retrieval.
The robustness that is evaluated is: 1) synthetic data distortions of the images and 2) text distortions (misspellings, typos)
The paper evaluates a set of recent models against this benchmark.

**Audience:**

No

**Claims And Evidence:**

Yes

**Requested Changes:**

Validation that evaluating on this benchmark has actual practical use in advancing either the science around the question, or a practical application.

**Strengths And Weaknesses:**

Strengths:
- Comprehensive set of benchmarks for a unique setting,
- Promise of an open benchmark.

Weaknesses:
I can not convince myself that this benchmark would actually be useful. Assume you have an application looking to perform text-image composed retrieval. You presumably want to do well in clean conditions, but also maybe to pay some cost on clean data to guarantee some non-catastrophic behavior on corrupted data.

What this paper suggests is that, on this problem, for all the pairs of models evaluated, the conclusion would be largely the same: picking the model that performs best on clean data is your best bet at getting a model robust to disturbances as well (the correlation is perfect on corrupted image data, almost perfect on text disturbances, enough that on balance it doesn’t matter)

Another scenario is that you have a model, and would want to produce a more robust variant of it to distortions seen in the wild. In that scenario, matching the kind of distortions seen in the wild, and in particular their distribution, is critical. This benchmark achieves neither: the distortions are synthetic, and their distribution is picked arbitrarily, and not reflective of their rate of appearing 'in the wild'.

My conclusion is that, by its very design, this benchmark adds more noise to the question of how to evaluate Text-Image Composed Retrieval than it adds signal. This is corroborated by the fact that no actionable findings are derived from this new benchmark in the paper.

I’m happy to hear a different perspective from the authors, preferably in the form of a section in the paper that validates that this benchmark has practical utility.

Nits: a number of the citations are rendered without proper spacing. ‘donated’ -> ‘denoted’.

---

### Review · Reviewer_i3ER · 2023-12-22

**Summary Of Contributions:**

This paper analyzes the robustness of various models against different types of corruptions in image and text domains on text-image composed retrieval tasks. This paper introduces three new datasets which are created by applying several image and text corruptions upon CIRR and FashionIQ for the analyses. The experimental results revealed conditions where each modality helps and hurts and suggested potential ways for improvements for the problem.

**Audience:**

Yes

**Broader Impact Concerns:**

No concern in this regard.

**Claims And Evidence:**

No

**Requested Changes:**

I would like to see a significant improvement to the writing. It would be possibly done by having a proofreading by a third-party person, but it has to improve not only text but also other entities such as tables and figures. For example, I failed to get the meaning of this sentence in Abstract:

"However, the robustness of these approaches against real-world corruptions or further text understanding has never been studied."

There are several texts like this and it makes it difficult to understand the content of the paper.

It would be very helpful to be able to know the major differences among methods in a table in terms of the comparing points such as the type of backbones.

The overall presentations of the figures also need to be improved. For example, using three terms, "top", "middle", "bottom" to classify five rows in Figure 2 is not very reader-friendly.

**Strengths And Weaknesses:**

Strengths:

1. This paper evaluates many methods on various conditions in a fair setting. This helps to understand the strengths and weaknesses of each method and discuss possible ways to improve text-image composed retrieval. The findings in the paper are not necessarily surprising but it is valuable to confirm those by rigid experiments.

2. This paper introduces new datasets and those will be shared publicly. The testing framework will also be shared. Those will help to make progress research in the field.

Weaknesses:

1. Presentation

The presentation of the paper needs to be significantly improved. The contents seem OK, but I find it very difficult to follow the text. It seems to be because of a combination of unpolished writing and less-than-optimal creations of tables and figures. The important details such as the formal definition of the task are not in the paper. There are many contents in appendix, but most of them is not mentioned in the paper.

2. Some details are not clear

Largely because of the problem mentioned above, I had difficulty to understand some of the experimental results.

My understanding is that text is to add information to image. If so, why text-only can even work reasonably? It could be obvious to researchers working on the particular datasets but at least it does not sound reasonable according to what is written in this paper.

In table 3, the results are better with different texts. Does it make sense? Are the comparisons between numbers meaningful? It was not clear what were actually compared.

---

> ### Author Response · Authors · 2024-03-03
> **Reply to Reviewer i3ER**
>
> We apologize for the unsatisfactory quality of our manuscript. After spending a considerable amount of time on it, we have since made significant efforts to enhance both language proficiency and readability. Furthermore, we have enlisted the assistance of our colleague, a native English speaker, for language corrections. We sincerely hope that these revisions have substantially improved the flow and language quality of the manuscript. In addition, we have highlighted the polished sections in red for easy identification.

---

> > ### Comment · Reviewer_i3ER · 2024-03-04
> > **Thank you for the response**
> >
> > I appreciate your effort to improve the quality of the manuscript. Indeed, it became much better in terms of readability. Now, I can focus more on the content of the paper. There are a couple of points that could make the paper better:
> >
> > 1. Make the claim of the paper clearer and write the paper to support the claim
> >
> > It says having robustness benchmark is important and provides new datasets for it. It is great, but it does not seem to bring a lot of insights as pointed by 6xaX. It makes the claim of the paper less clear. Actually, I find it difficult to understand what is being claimed by the paper by reading the abstract and introduction beyond that it provides new datasets and give experimental results on those. I would like to be able to understand the claim of the paper more easily and the claim is supported by the experiments.
> >
> > 2. The experimental design may not be adequate
> >
> > I really appreciate the number of experiments conducted in the paper and it could provide some insights to the community. However, it seems to need more work to have a deeper discussion the results.
> >
> > The robustness is measured by a metric that computes a relative reduction of a metric in a noisy condition compared to the clean condition. It should be a relevant metric, but the metric alone could not be sufficient for discussing the robustness of a method. Without discussing the absolute number, it may result in a wrong conclusion. For example, if a method always produces a wrong answer, it needs to be considered as a metric which has the highest robustness (of course, 0/0 is undefined and the metric may not say it is the best, but I hope what I mean here is clear).  The paper reports the absolute numbers, too, but the discussion on robustness seems to more or less ignore it.
> >
> > The benchmark for the text understanding capability seems valuable, but I am not sure how to interpret the experimental results. Given that the test data are different, conclusions we could draw from the results are weak. At least, we cannot discuss the robustness of a method comparing the absolute numbers on different sets. We could discuss relative differences among methods, but not capabilities.

---

> > > ### Author Response · Authors · 2024-03-04
> > > **Reply to reviewer i3ER**
> > >
> > > Thank you for your time and helpful feedback. We respond below to your questions and concerns:
> > > >Make the claim of the paper clearer and write the paper to support the claim
> > >
> > > We would like to point out that our claims are presented in italics after each paragraph in Section 4, and we provide suggestions in the conclusion listed as below:
> > > 1) Model pretrained on large datasets will lead to better robustness,
> > > 2) Text features from aligned space can help boost the robustness, while text features from independent space will reduce robustness
> > > 3) A modified text is more likely to enhance the model’s discriminative ability when it minimizes the number of feasible targets and will harm performance when producing more candidate responses.
> > >
> > > We have updated the introduction with the aforementioned conclusion to enhance clarity.
> > > Additionally, there is currently no research within the community on the robustness aspect of text-image composed retrieval. We pioneered a testbed for evaluating the robustness of models in the text-image composed retrieval task, and our claims are supported by the experimental results from Table 2, 3, and 4.
> > >
> > > > The paper should reports the absolute numbers, too
> > >
> > > The clean recall performance is reported in the first column of Table 2 and Table 3. The relation between relative robustness and recall performance is plotted in Figure 4.
> > >
> > > > how to interpret the experimental results for text understanding.
> > >
> > > We didn’t directly compare the absolute numbers on different sets, instead, we compared the variations of each set with the first column, which is a collection of different text queries involving variations in numerical aspects, attributes, object removal, and background. Both the collection and single categories are retrieved from the same gallery.
> > >
> > > By comparing the performance of the collection query and each category, we observed that the removal and background categories yielded lower score across most of the models. Furthermore, when examining the variations for different models within the same text category, we noticed that two Image-only models showed improvement, whereas models incorporating text experienced a decrease in performance. Consequently, we speculate that the inclusion of text within these removal/background categories may negatively impact performance. We further claim that the modified text will harm performance when producing more candidate responses.

---

### Review · Reviewer_Cb81 · 2024-02-05

**Summary Of Contributions:**

This paper introduces a robustness benchmark for the task of composed image retrieval. For each of the two CIR datasets (CIRR and FashionIQ), the authors propose:
1. Corruptions for the visual part of the dataset in the style of Hendrycks and Dietterich 2019
2. Corruptions from the textual part of the dataset in the style of Rychalska et al. 2019

The authors also propose a diagnostic dataset for textual understanding based on instances of the CIRR dataset and synthetic data.

**Audience:**

Yes

**Claims And Evidence:**

No

**Requested Changes:**

Critical for acceptance:
1. In order to improve W1 and W3-2: Add pic2word and SEARLE results to tables 2 and 3. This will check if your conclusion 2 holds for more CLIP-based methods
2. In order to improve W3-1: Remove “with little distribution shift” from conclusion 1 or find a way to motivate it.
3. In order to improve W3-1: Design an experiment (choose one of your datasets eg CIRR-C) to see if the robustness is caused by a) CLIP architecture, b) number of weights, c) pretraining. One suggestion (you could choose something else) is to use the SEARLE-OTI method that is zero shot and also a test-time optimization method and directly apply it to open-clip i) vit-L/14 LAION-2B, ii) vit-H/14 LAION-2B iii) vit-L/14 OpenAI's WIT (or DataComp-1B) so you can study the role of weights on the robustness (comparison i vs ii) and the role of the pretraining (comparison i vs iii).
4. In order to improve both W3-1 and W3-2: design an experiment with other Language-Image models like blip/blip2. The method you choose to demonstrate is not important to me but I suggest using SEARLE-OTI because it is the easiest to try. This will check if your conclusion 2 holds for non-CLIP-based methods.


Simply strengthen the work:
1.  In order to improve W2: Add a CIRCO-C subtable in table 2 with image-only (CLIP), text-only (CLIP), pic2word, SEARLE
2. I suggest removing the standard deviations from figure 3, given that they are overlapping in the level of being unreadable. Also the right subfigure of figure 3 could be on a different double figure e.g. figure x, with figure x (a) circle size to be the number of model parameters and figure x (b) size to be the number of pretraining dataset.

**Strengths And Weaknesses:**

The novelty of this work is limited, in the sense that the two corruption approaches (visual and textual) are the same with the literature cited in the paper (Hendrycks and Dietterich 2019, Rychalska et al. 2019). On the other hand, this work can potentially be useful to the composed image retrieval community with some improvements.

W1. This work does not include the latest methods of composed image retrieval, namely pic2word [1], SEARLE [2]. Both methods were published before the submission of this paper and they have code (and models) online.

W2. Given that this paper does not introduce novel corruptions, it can increase its value by performing more experiments/comparisons so a third dataset like CIRCO is suggested to be used.

W3. Conclusions are not derived from the experiments.
1. “Models pretrained on larger datasets with little distribution shift will lead to better robustness”: First, we cannot know what kind of distribution shifts exist in a 400m dataset except if we investigate manually. Someone could argue that the robustness of those models exist specifically because the distribution shifts are more diverse in those datasets. Second, from the experiments we can see that the robustness correlates with both dataset size and the number of model parameters (Figure 3 right). So claiming that the reason for robustness is the pretraining is not supported by the experiments.
2. “Text features from an aligned space can help boost the robustness, while text features from independent space will damage the model robustness”: Only one aligned space is used for this experiment so the evidence of this general statement is not strong.

[1] Saito, Kuniaki and Sohn, Kihyuk and Zhang, Xiang and Li, Chun-Liang and Lee, Chen-Yu and Saenko, Kate and Pfister, Tomas. Pic2Word: Mapping Pictures to Words for Zero-shot Composed Image Retrieval. CVPR 2023

[2] lberto Baldrati and Lorenzo Agnolucci and Marco Bertini and Alberto Del Bimbo. Zero-Shot Composed Image Retrieval with Textual Inversion. ICCV 2023

---

> ### Author Response · Authors · 2024-02-19
> **Reply to Reviewer Cb81**
>
> Thank you for your time and helpful feedback. We respond below to your questions and concerns:
>
> > W1 and W3-2:  This work does not include the latest methods of composed image retrieval, namely pic2word [1], SEARLE [2]. Only one aligned space is used for this experiment so the evidence of this general statement is not strong.
>
> We followed the suggestion and added pic2word and SEARLE results to Tables 2 and 3. Both methods are based on CLIP and show higher robustness under various image corruptions, 0.83 for Pic2word and 0.84 for SEARLE on average.
>
> Furthermore,  we trained BLIP2 for or task following the CLIP4CIR method, denoted as BLIP2-CIR. The BLIP2 is pretrained with a text-image aligned space similar to CLIP. The results show that the average relative robustness of BLIP2-CIR is 0.9 among 15 visual corruptions.
>
> Both sets of the results support our second conclusion: Text features from an aligned space can help enhance robustness
>
> > W2: Given that this paper does not introduce novel corruptions, it can increase its value by performing more experiments/comparisons so a third dataset like CIRCO is suggested to be used.
>
> We followed the suggestion and evaluated Image-only, Text-only, Pic2word and SEARLE models on CIRCO-C under various corruptions. The results are shown in Table 2. On average, Pic2word and SEARLE exhibit higher robustness compared to Image-only and Text-only methods.
>
> > W3-1: “Models pretrained on larger datasets with little distribution shift will lead to better robustness”: First, we cannot know what kind of distribution shifts exist in a 400m dataset except if we investigate manually.
>
> For this conclusion, we mean that text-image aligned pretrained , such as CLIP, provide better robustness. Separate pretraining, such as using ResNet as the image encoder and LSTM as the text encoder, results in lower performance. This is also reflected in the second conclusion, so we remove the phrase 'with little distribution shift' here to avoid potential confusion.
>
> > The experiments we can see that the robustness correlates with both dataset size and the number of model parameters (Figure 3 right). So claiming that the reason for robustness is the pretraining is not supported by the experiments.
>
> We followed the suggestion and trained three CLIP4CIR models with i) vit-L/14 LAION-2B, ii) vit-H/14 LAION-2B iii) vit-L/14 LAION-400M. The results demonstrate that both the size of the pretrained dataset and the size of model parameters contribute to higher robustness, with the size of model parameters playing a more significant role. Specifically, changing the model structure from ViT-L14 (487.9M) to ViT-H14(1049M) can lead to a 3% improvement in relative robustness. Changing the size of the pretrained dataset from LAION-400M to LAION-2B results in a 1% improvement in relative robustness.
>
> > right subfigure of figure 3 could be on a different double figure.
>
> Thanks for the suggestion, we already make change in the updated version.

---

> > ### Comment · Reviewer_Cb81 · 2024-03-01
> >
> > I really appreciate your effort answering my concerns and in this stage I am already positive about this work. I am completely satisfied about the quality and the number of your experiments. I tend to agree with the reviewer i3ER about the poor manuscript, if you could improve that I would be fully positive about this paper.

---

> ### Author Response · Authors · 2024-03-03
> **Reply to Reviewer Cb81**
>
> We apologize for the unsatisfactory quality of our manuscript. After spending a considerable amount of time on it, we have since made significant efforts to enhance both language proficiency and readability. Furthermore, we have enlisted the assistance of our colleague, a native English speaker, for language corrections. We sincerely hope that these revisions have substantially improved the flow and language quality of the manuscript. In addition, we have highlighted the polished sections in red for easy identification.

---

### Decision · Action_Editor_YZVx · 2024-03-13

**Recommendation:** Reject

**Comment:**

In their final recommendations, the reviewers all acknowledge the interest of better benchmarking composed retrieval. However, they express concerns regarding:
- the limited realism of the proposed benchmark,
- the lack conclusions that one can draw from this benchmark,
- the lack actionable items that can be identified from the the authors' results,
- the clarity and writing quality of the paper (although this was improved by the authors).
Ultimately, two out of three reviewers express doubts that this benchmark will benefit the community and see risks that in adds further noise in the benchmarking of composed retrieval models.

**Audience:**

No. As stated above, the reviewers see the potential of a benchmark for composed retrieval but are not truly convinced that the benchmark proposed in this work fulfills a need in the community.

**Claims And Evidence:**

No. Although the reviewers acknowledge the interest of having benchmarks for composed retrieval, they express concerns in terms of the conclusions that can be drawn from evaluation on the proposed benchmark and the lack of clear actionable items based on the evaluation performed by the authors.